# Structural and mechanistic characterization of bifunctional heparan sulfate N-deacetylase-N-sulfotransferase 1

Courtney J. Mycroft-West[1], Sahar Abdelkarim[1], Helen M. E. Duyvesteyn[2], Neha S. Gandhi [3,4,5], Mark A. Skidmore [6], Raymond J. Owens [1,2] & Liang Wu [1,2] ✉

Heparan sulfate (HS) polysaccharides are major constituents of the extracellular matrix, which are involved in myriad structural and signaling processes. Mature HS polysaccharides contain complex, non-templated patterns of sulfation and epimerization, which mediate interactions with diverse protein partners. Complex HS modifications form around initial clusters of glucosamine-N-sulfate (GlcNS) on nascent polysaccharide chains, but the mechanistic basis underpinning incorporation of GlcNS itself into HS remains unclear. Here, we determine cryo-electron microscopy structures of human N-deacetylase-N-sulfotransferase (NDST)1, the bifunctional enzyme primarily responsible for initial GlcNS modification of HS. Our structures reveal the architecture of both NDST1 deacetylase and sulfotransferase catalytic domains, alongside a non-catalytic N-terminal domain. The two catalytic domains of NDST1 adopt a distinct back-to-back topology that limits direct cooperativity. Binding analyses, aided by activity-modulating nanobodies, suggest that anchoring of the substrate at the sulfotransferase domain initiates the NDST1 catalytic cycle, providing a plausible mechanism for cooperativity despite spatial domain separation. Our data shed light on key determinants of NDST1 activity, and describe tools to probe NDST1 function in vitro and in vivo.

Heparan sulfate (HS) is a ubiquitous and evolutionarily ancient class of polysaccharide produced throughout the metazoan lineage[1]. HS polysaccharides are abundant within the extracellular matrix (ECM) of multicellular organisms in the form of HS proteoglycans (HSPGs), wherein one or more polysaccharide chains are covalently linked to a membrane bound or pericellular core protein[2]. HSPGs in the ECM interact with myriad partners, and are essential regulators of physiological processes including growth factor signaling[3], morphogen patterning[4], adhesion[5], endocytosis[6], barrier filtration[7] and host-pathogen binding[8–10].

HS is a member of the glycosaminoglycan (GAG) family—linear polysaccharides typically comprised of alternating hexosamine and uronic acid monosaccharides (heparin, HS, chondroitin sulfate, dermatan sulfate and hyaluronan), whilst keratan sulfate is comprised of alternating hexosamine and galactose units. The hexosamine sugars of HS are predominantly N-acetyl- or N-sulfo-glucosamine (GlcNAc,

[1]The Rosalind Franklin Institute, Harwell Science & Innovation Campus, OX11 0QX Didcot, UK. [2]Division of Structural Biology, Nuffield Department of Medicine, University of Oxford, The Wellcome Centre for Human Genetics, OX3 7BN Oxford, UK. [3]Department of Computer Science and Engineering, Manipal Institute of Technology, Manipal Academy of Higher Education, Manipal 576104 Karnataka, India. [4]School of Chemistry and Physics, Queensland University of Technology, QLD 4000 Brisbane, Australia. [5]Centre for Genomics and Personalised Health, Queensland University of Technology, Brisbane QLD 4059, Australia. [6]Centre for Glycoscience Research and Training, Keele University, ST5 5BG Newcastle-Under-Lyme, UK. ✉e-mail: liang.wu@rfi.ac.uk

GlcNS respectively), and the uronic acids of HS are glucuronic or iduronic acid (GlcA, IdoA). Each monosaccharide unit of HS can be further elaborated by variable O-sulfation (Supplementary Fig. 1a)[11], giving rise to diverse polysaccharide compositions that enable binding to multiple partners[12]. HS sequence complexity is crucial for its biological function, and is tightly regulated by cells in response to developmental cues, physiological status and external stimuli[13].

HS biosynthesis begins with polymerization to produce unmodified [GlcNAc-GlcA]$_n$ chains (hereafter heparosan) by the *EXT* family of glycosyltransferases[14–16]. Complexity is introduced by a series of modification reactions upon the growing heparosan chain (Supplementary Fig. 1b), creating a modular polysaccharide structure in which NS domains, containing high proportions of IdoA and sulfated sugars, are flanked by NA/NS transition domains of intermediate modification, and unaltered NA domains that resemble unmodified heparosan. The biological functions of HS are dictated by the presentation of highly modified NS domains, or adjoining NS/NA regions, which predominantly participate in protein interactions[2].

Bifunctional N-deacetylase-N-sulfotransferase (NDST) enzymes are the first to act on nascent heparosan after its polymerization, converting GlcNAc sugars to GlcNS, via deacetylated glucosamine (GlcN) intermediates (Supplementary Fig. 1c). HS modification by NDSTs does not occur stochastically, but instead produces discrete GlcNS clusters of variable length[17,18]. Notably, these GlcNS clusters are themselves recognized by downstream enzymes, which work to generate further HS modifications, ultimately producing the highly sulfated and epimerized sequences characteristic of NS domains. The activity of NDST enzymes thus plays a central role in determining the overall locations and extents of modification within HS polysaccharide chains (Fig. 1A)[13,19–21].

Humans express 4 members of the NDST family (NDST1–4). NDST1 and NDST2 are systemically distributed, with NDST1 dominant in most tissues, and NDST2 dominant in mast cells, which produce the highly modified HS analogue heparin[22,23]. NDST3 and NDST4 show more limited distribution, with both isoforms abundant in the brain[24–26]. Commensurate with their essential role in HS biosynthesis, dysregulation or mutations of NDSTs are associated with a spectrum of pathologies. Total loss of Ndst1 produces perinatal lethality in mice due to severe cerebral and craniofacial defects[27], with impaired lung development also noted[28,29]. Partially inactivating *NDST1* mutations are also linked to autosomal recessive intellectual disability in humans[30], whilst changes to *NDST3* expression have been implicated in schizophrenia and bipolar disorder[31], and loss of NDST4 has been suggested to be a prognostic marker for adverse colorectal cancers[32].

Despite substantial interest, the molecular basis of NDST activity remains poorly understood. An X-ray crystal structure of the NDST1 sulfotransferase domain was reported in 1999 (PDB 1NST)[33], providing some insight into the basis of sulfate transfer from 5′-phosphoadenosine-3′-phosphosulfate (PAPS) to deacetylated GlcN substrates. However, little is known about NDST deacetylase activity, and consequently, about how the deacetylase and sulfotransferase activities of NDSTs are coordinated. Biochemical experiments have demonstrated reduced GlcNS formation and no GlcNS clustering when heparosan oligosaccharides are modified by a mixture of truncated NDST1 deacetylase and sulfotransferase domains, strongly implying functional cooperativity in the full-length enzyme[18]. However, a lack of structural information has precluded analysis of the basis for such functional coupling to date.

Here, we present high-resolution cryogenic-electron microscopy (cryo-EM) structures of the major human NDST isoform NDST1. Structural characterization, aided by activity modulating nanobodies, reveal important determinants of NDST1 catalytic function, including active site architectures, and key residues and loops involved in substrate processing. In contrast to the known bifunctionality of NDST1, our structures clearly reveal a 3-domain enzyme, with the sulfotransferase and deacetylase domains flanked by a non-catalytic N-

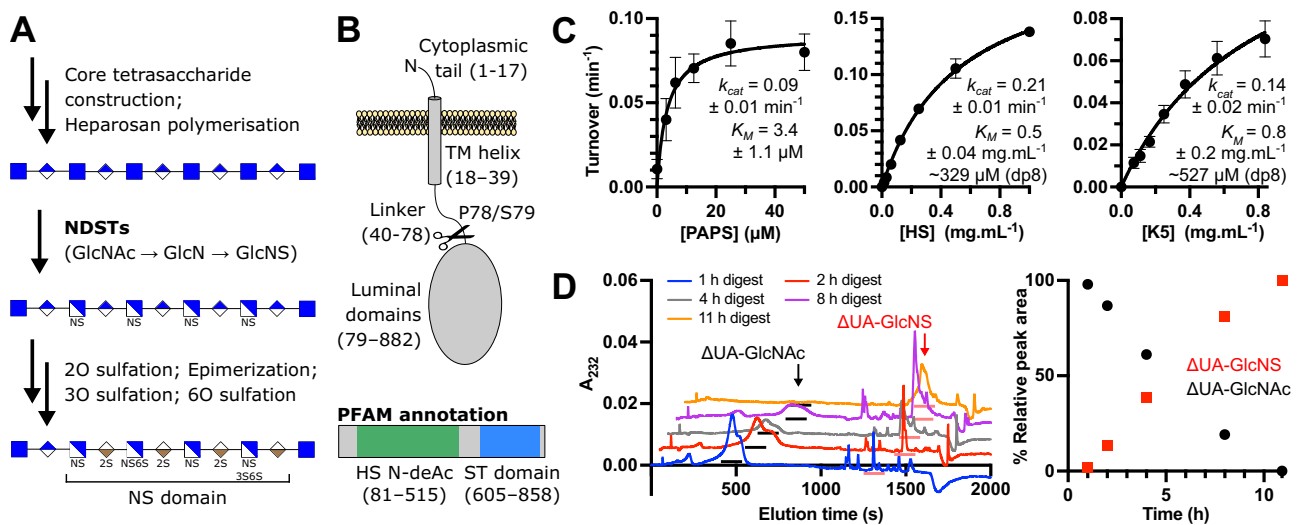

**Fig. 1 | Activity profile of NDST1. A** Sequential pathway for HS biosynthesis. The reaction catalyzed by NDST enzymes (conversion of GlcNAc to GlcNS via glucosamine) is highlighted in bold. GlcNS sugars produced by NDSTs are recognized and targeted by downstream enzymes to create highly modified NS domains. **B** Schematic diagram of NDST1 on the Golgi membrane, and the site of truncation to make solubilized recombinant NDST1. PFAM[73] annotation of NDST1 is shown, depicting predicted deacetylase (HS N-deAc) and sulfotransferase (ST) domains. **C** Pseudo-first order Michaelis-Menten kinetics for NDST1 with respect to sulfate donor substrate PAPS, and acceptor polysaccharide substrates HS and K5. Datapoints are mean ± s.d. for $N = 3$ (HS and PAPS) or 4 (K5) technical replicates run on the same assay plate. Uncertainties for kinetic constants represent standard errors

of curve fitting. Source data are provided within the Source Data file.
**D** Disaccharide analysis of NDST1 activity on K5 polysaccharide, showing quantitative conversion of GlcNAc residues to GlcNS. Analysis was run in singlicate ($N = 1$). Left—chromatographs of disaccharides generated from K5 after treatment with NDST1 for the indicated timepoints (staggered for clarity). Peaks corresponding to ΔUA-GlcNAc and ΔUA-GlcNS are annotated by arrows, and underscores below the peaks. Minor unannotated peaks are impurities from the reaction mixture. Right—Conversion of GlcNAc to GlcNS over time. Datapoints represent peak areas normalized to the total area of both ΔUA-GlcNAc and ΔUA-GlcNS peaks. Source data are provided within the Source Data file.

terminal domain. Counterintuitively, we find that the deacetylase and sulfotransferase catalytic domains of NDST1 project in opposing directions, limiting possible mechanisms of coordination. Analysis of NDST1-HS binding suggests that initial anchoring interactions likely occur at the sulfotransferase domain, despite deacetylation taking place first during the NDST1 catalytic cycle. Based on these observations, we propose a model for NDST1 bifunctionality that can operate even in the absence of direct domain-to-domain transfer.

## Results

### Production and characterization of recombinant NDST1

NDST1 is a Golgi lumen resident type-II transmembrane protein, which contains a single pass N-terminal helix that anchors the catalytic luminal portions of the enzyme to the Golgi membrane. To facilitate structural and biochemical characterization, we generated solubilized NDST1 truncated at P78/S79, removing the N-terminal helix and a linker region (Fig. 1B). Solubilized recombinant NDST1 was expressed using a baculoviral expression system and purified to homogeneity via metal ion affinity, anion exchange, and size-exclusion chromatography.

We tested the in vitro enzymatic activity of solubilized NDST1 using a recently described coupled enzyme system, in which NDST1 activity is linked to the rat sulfotransferase Sult1A1, which catalytically regenerates sulfate donor PAPS from fluorogenic 4-methylumbelliferyl (4-MU) sulfate[34] (Supplementary Fig. 2a, b). As deacetylation by NDST1 is typically rate limiting[25], depending on substrate concentrations, this coupled enzyme assay primarily probes this step of turnover. Initial activity trials were carried out using $Mg^{2+}$ supplemented in the assay buffer, to satisfy the requirement of NDST1 catalysis on divalent cations, which are necessary for deacetylation. Surprisingly, we also noted robust activity in the absence of divalent supplementation, which was inhibited by treatment with the chelators EDTA or dipicolinic acid (DPA) (Supplementary Fig. 2c). Inductively coupled plasma optical emission spectroscopy (ICP-OES) revealed that this was the result of residual NDST1 associated cations, primarily $Ca^{2+}$ with trace $Zn^{2+}$, carried over from protein purification. Further supplementation with $Ca^{2+}$ did not improve activity, indicating that carryover was stoichiometric (Supplementary Fig. 2c, d). Prior analysis by Dou et al. suggests that NDST1 deacetylation is compatible with a range of metals, including $Mg^{2+}$, $Mn^{2+}$, $Ca^{2+}$ and $Co^{2+}$, with highest activity found in the presence of $Ca^{2+}$ [18]. We thus assigned the most likely identity of the metal cofactor in our NDST1 samples to be $Ca^{2+}$.

Pseudo-first order activity constants were determined for NDST1 with respect to PAPS ($k_{cat} = 0.09$ $min^{-1}$; $K_M = 3.4$ μM), as well as sulfate acceptors HS ($k_{cat} = 0.21$ $min^{-1}$; $K_M = 0.5$ mg $mL^{-1}$) and E. Coli K5 – a capsular polysaccharide that shares the same repeating disaccharide motif as heparosan[35] ($k_{cat} = 0.14$ $min^{-1}$; $K_M = 0.8$ mg $mL^{-1}$; Fig. 1C). NDST1 has previously been proposed to act upon 8-mer stretches of heparosan (MW 1519.3 Da)[18], thus these $K_M$ constants for HS and K5 approximate to ~329 and ~527 μM respectively.

To further confirm the activity of solubilized NDST1, we followed the conversion of GlcNAc to GlcNS in K5 polysaccharide by disaccharide analysis, employing exhaustive digestion by heparin lyase II from P. heparinus. NDST1 treatment of K5 resulted in time-dependent loss of ΔUA-GlcNAc peaks, matched to increasing ΔUA-GlcNS, clearly demonstrating enzymatic activity in a bona fide polysaccharide context (Fig. 1D).

### Activity modulating nanobodies for NDST1

To aid mechanistic investigations of NDST1, we generated de novo nanobody (nAb) binders, which can provide insight by trapping conformational states relevant to function. A panel of NDST1 binding nAbs were isolated by primary immunization of a llama host with recombinant solubilized protein, followed by phage display bio-panning of the resulting immunized $V_{HH}$ library over two rounds[36].

Binding analysis using surface plasmon resonance (SPR) and biolayer interferometry (BLI) confirmed 5 NDST1 interacting nAbs. Binding affinities ($K_D$s) spanned the high nanomolar to the low micromolar range, in the rank order nAb7 > nAb13 ≈ nAb5 ≈ nAb6 ≫ nAb1, with rapid on/off kinetics in all cases (Fig. 2A, Supplementary Fig. 3, Table 1). Additive SPR signals indicative of co-binding were detected for most pairwise combinations of nAbs, except nAb5 + nAb7, nAb5 + nAb13 and nAb7 + nAb13, which displayed mutually exclusive binding (Supplementary Fig. 4).

We also assessed nAbs for their ability to modulate NDST1 activity using the coupled enzyme assay. NDST1 inhibition was observed in the presence of nAb5 and nAb7, suggesting that these nAbs bind regions relevant for catalysis. Unexpectedly, we also observed an approximately 2-fold enhancement of NDST1 activity upon nAb13 binding, indicative of allosteric modulation that enhances catalytic turnover (Fig. 2B, Table 1). Differential scanning fluorimetry showed ~2.6 °C, ~4.4 °C and ~1.9 °C stabilization of NDST1 thermal denaturation in the presence of 10.8 μM nAb5, nAb7 and nAb13 respectively, broadly in line with their relative binding affinities for the enzyme (Supplementary Fig. 5).

### Cryo-EM structure of bifunctional NDST1

To better understand the molecular basis for NDST1 activity, and how it is modulated by nAb binding, we sought to resolve three-dimensional structures of NDST1 and its complexes with nAb7 and nAb13 (Supplementary Fig. 6). A cryo-EM volume of NDST1 alone was calculated to a nominal resolution of 2.70 Å (Relion Au-FSC = 0.143), with the resulting map of sufficient quality to model nearly all residues along the protein chain, using the AlphaFold2 model as a starting point for refinement (Fig. 3A; Supplementary Fig. 7; Supplementary Table 1).

NDST1 adopts an elbow-shaped monomeric structure comprised of 3 clearly resolved domains, in contrast to its widespread annotation as a bifunctional protein (Fig. 3B; compare Fig. 1B). Disulfide bonds between C586–C601 and C818–C828 were clearly visible in the cryo-EM density, with the former at lower occupancy, possibly due to its surface location being susceptible to reduction by 1 mM DTT, present in the sample buffer (Supplementary Fig. 8). Proteins produced by baculoviral expression can sometimes differ in glycosylation compared to mammalian expression[37]. However, we were unable to resolve distinct glycoforms in our NDST1 sample by cryo-EM. Density consistent with the first GlcNAc of an N-glycan tree could be seen at the predicted N-glycosylation site N401, but this was not of sufficient quality to model (Supplementary Fig. 9).

The previously solved NDST1 sulfotransferase domain resides at the C-terminus of the protein (residues D602–R882), and our cryo-EM model agrees well with the crystal structure (PDB accession 1NST; RMSD 0.96 Å over 277 Cαs; Supplementary Fig. 10a)[33], displaying a well-defined binding cleft capable of accommodating HS polysaccharides (Fig. 3C). We observed unambiguous density within the sulfotransferase active site corresponding to a molecule of 5'-phosphoadenosine-3'-phosphate (PAP; present in the cryo-EM sample), which binds NDST1 via H-bonds and salt bridges to K614–T618, R782, K833, R835, Y837, R782 and W817, as well as a π-stacking interaction from adenine to F816 (Fig. 3D). The C828–D841 loop at the mouth of the PAP site partially occludes the ligand from solvent, suggesting a role as a stabilizing lid that encloses PAP(S) after its engagement with the sulfotransferase domain (Supplementary Fig. 10b). The positions of both PAP and the C828–D841 loop closely match the previously solved crystal structure.

Deacetylase activity was assigned to the central domain of NDST1 (L308–C601), based on close structural homology to the fungal galactosaminogalactan deacetylase Agd3 from Aspergillus fumigatus (PDB accession 6NWZ; 21.5% identity to NDST1 residues 1-601; Supplementary Fig. 11a, b)[38]. The deacetylase domain of NDST1 adopts a $(\beta/\alpha)_7$ barrel fold with a clear HS binding cleft (Fig. 3C), and contains a

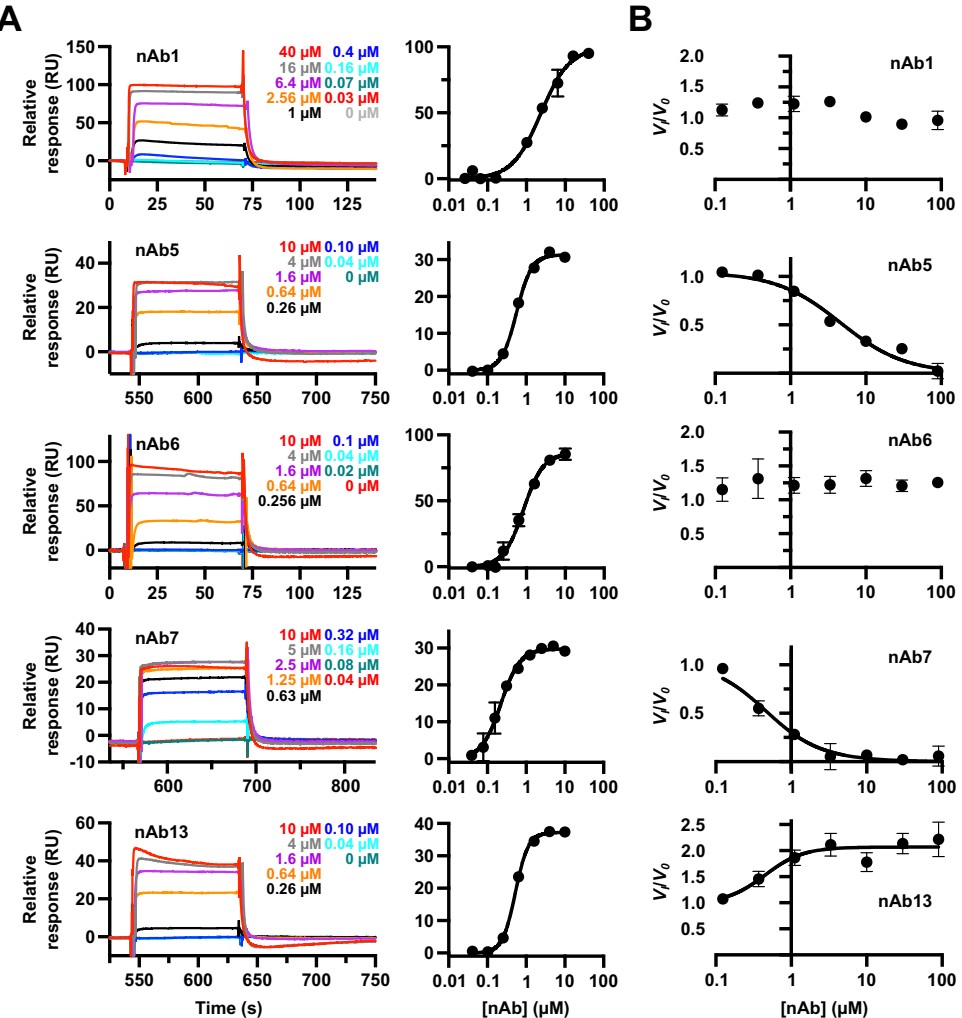

**Fig. 2 | Biophysical characterization of anti-NDST1 nAbs. A** Representative SPR sensorgrams and steady state binding curves showing interaction of nAbs with surface bound NDST1. Sensorgrams were collected in technical triplicate ($N = 3$), with similar results observed for all replicates. Datapoints on binding curves are mean ± s.d. for $N = 3$ technical replicates. RU−response unit. **B** NDST1 activity modulation in the presence of nAbs. nAb5 and nAb7 are inhibitory, whilst nAb13 enhances NDST1 activity. Datapoints are mean ± s.d. for $N = 3$ technical replicates run on the same assay plate. Quantitated binding affinities and kinetic parameters are summarized in Table 1. Source data for all plots are provided within the Source Data file.

His-His-Asp triad characteristic of metal-dependent deacetylases[39]. Clear density, distinct from surrounding protein sidechains, was visible at the center of this triad, consistent with $Ca^{2+}$ coordinated to H389, H393, D320, although a full coordination sphere could not be resolved, possibly due to local disorder within the open catalytic cleft (Fig. 3E). Further density within the deacetylase active site was tentatively modeled as a second $Ca^{2+}$, consistent with the presence of multiple divalent ions in the active site of Agd3 and other deacetylases[38,40] (Supplementary Fig. 11c). To confirm the role of the deacetylase domain active site residues, we tested the activity of H389A, H393A and D320A mutants, as well as H529A and D319A – two nearby residues that likely function as general acid/base during deacetylation (see below). In all cases, mutation to alanine abrogated NDST1 turnover with minimal disruption to protein structure, demonstrating key roles for these sidechains in enzymatic processing (Fig. 3F; Supplementary Fig. 12).

To our knowledge, the NDST1 N-terminal domain (NTD; residues 79−307) has not been previously reported. The NDST1 NTD adopts a Rossman-like fold, comprising a central layer of β-sheets flanked by α-helices on each side, although disorder prevented modeling of loops K171−S179, S191−D197, V215−W225 and K242−L262 (Fig. 3C). Interestingly, a similar non-catalytic domain in Agd3 was proposed to function as a carbohydrate-binding module, which extends the span of the substrate binding cleft[38]. Whilst the analogous cleft in NDST1 is occluded by the Q95−S106 helix (Supplementary Fig. 11d), inspection of the NDST1 surface does reveal a channel formed at the deacetylase and N-terminal domain interface, which extends the deacetylase cleft via a sharp turn (Supplementary Fig. 11e). This channel may guide long HS polysaccharides out of the deacetylase active site, and implicates a role for the NDST1 NTD in controlling enzyme-substrate interactions.

## Structures of NDST1-nAb complexes

Complexes of NDST1 with activity-modulating nanobodies nAb7 and nAb13 were isolated by size-exclusion chromatography, and the resulting purified complexes analyzed by cryo-EM (Supplementary Fig. 6).

Initial reconstructions of NDST1-nAb7 showed substantial disorder, consistent with unresolved heterogeneity, which was revealed by 3D variability analysis to correspond to a hinging motion around the NDST1 elbow (Supplementary Movie 1). Further local refinements were thus carried out, focussed around either side of the hinge, yielding 2.42 Å partial maps (Au-FSC = 0.143) that were combined for model building purposes (Supplementary Fig. 7; Supplementary Table 1). The resulting composite map was of sufficient quality to model both NDST1

**Table 1 | NDST1 binding and activity modulation by nAbs, as measured by SPR, BLI and enzyme kinetics**

| Nanobody | $K_D$ (SPR)/µM | $K_D$ (BLI)/µM | Activity modulation/µM |
|---|---|---|---|
| nAb1 | 2.37 ± 0.22 | 2.67 ± 0.57 | > 100 |
| nAb5 | 0.57 ± 0.02 | N.D. | 3.30 ± 0.37 ($K_I$ inhibition) |
| nAb6 | 0.84 ± 0.05 | 0.47 ± 0.16 | > 100 |
| nAb7 | 0.22 ± 0.01 | 0.12 ± 0.03 | 0.24 ± 0.04 ($K_I$ inhibition) |
| nAb13 | 0.53 ± 0.01 | 0.24 ± 0.13 | 0.45 ± 0.13 ($EC_{50}$ enhancement) |

SPR and BLI affinities calculated from steady state binding curves. N.D. not determinable due to experimental artifact. Uncertainties represent standard errors of curve fitting.

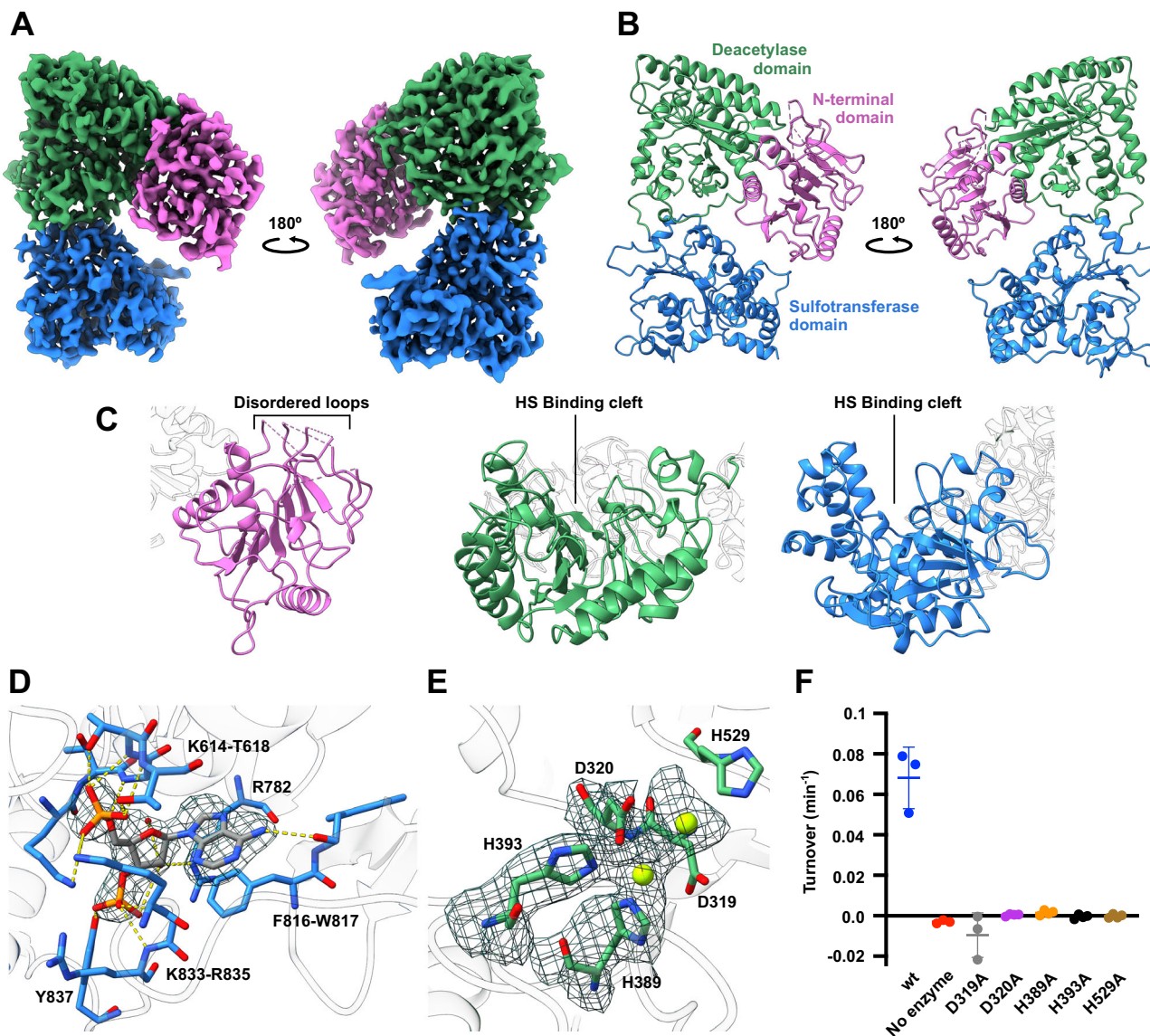

**Fig. 3 | Three-dimensional structure of NDST1. A** Cryo-EM coulombic density map of NDST1 contoured to 0.12 in ChimeraX, and colored by domain. **B** Ribbon model of NDST1 complex, colored by domain. **C** Close-up views of individual NDST1 domains, highlighting features of interest. Colors as in (**B**). **D** Molecular architecture of the NDST1 sulfotransferase active site, showing interactions between bound PAP and neighboring amino acids. H-bonds are depicted by yellow dashes. Red sphere is a water molecule. Density contoured to 0.20. **E** Molecular architecture of the NDST1 deacetylase active site, showing coordination of catalytic $Ca^{2+}$ ions (green spheres) around the D320, H393, H389 triad. Density contoured to 0.15. **F** Activity of NDST1 deacetylase domain mutants. Alanine mutation of residues at the catalytic center causes loss of activity. Datapoints are measurements from $N = 3$ (wt, no enzyme, D319A) or 4 (D320A, H389A, H393A, H529A) technical replicates run on the same assay plate. Bars show mean ± s.d. Source data are provided within the Source Data file.

and nAb7 in detail, including the amino acid sidechains present at their interface.

Examination of the NDST1-nAb7 complex revealed nAb binding at the NDST1 deacetylase domain, across its substrate cleft, providing a clear steric rationale for inhibition (Fig. 4A, B). nAb7 binding was predominantly mediated through its CDR3 loop, with H-bonds between $D593_{NDST1}$–$Y102_{nAb7}$, $K465_{NDST1}$–$Q113_{nAb7}$, $R590_{NDST1}$–$R99_{nAb7}$, $K589_{NDST1}$–$Y58_{nAb7}$ and salt bridges between $D593_{NDST1}$–$K101_{nAb7}$, $E459_{NDST1}$–$R99_{nAb7}$, $K465_{NDST1}$–$D99_{nAb7}$ and $K465_{NDST1}$–$D115_{nAb7}$ (Fig. 4C; Supplementary Fig. 13a). The overall structure of NDST1 complexed to nAb7 was similar to that of free NDST1, except for a slightly more closed conformation of the NDST1 elbow, consistent with conformational flexibility around its hinge (Supplementary Fig. 14; Supplementary Movie 1). We also observed improved ordering of the D319−V332, C486−G512 and T528−L538 helical loops, likely due to interactions with the proximal nAb7 (Supplementary Fig. 15).

The NDST1-nAb13 volume was reconstructed following the same local refinement strategy as NDST1-nAb7, owing to a similar hinging movement (Supplementary Movie 2; Supplementary Fig. 7; Supplementary Table 1). Consistent with its weaker binding, density for the nAb13 portion of the reconstructed map appeared disordered, and could only be modeled by rigid-body fitting of an all Cα nAb chain, derived from the nAb7 structure (Fig. 4D).

nAb13 binds toward the side of the deacetylase domain, in a region that does not obstruct the substrate cleft (Fig. 4E). The rigid-body fitted nAb13 chain was found to lie within 6 Å of NDST1 residues K331, E333, K336 and R537, indicating these amino acids may be involved in binding (Supplementary Fig. 13b). As with nAb7, the nAb13 complex displayed a more closed NDST1 conformation, as well as improved ordering of the D319-V332, C486-G512 and T528-L538 loops (Supplementary Fig. 14, 15). No overlap was observed between the nAb7 and nAb13 binding sites, suggesting their inability to mutually bind NDST1 arises from allosteric effects (Supplementary Fig. 4, 16).

We hypothesized that enhancement of NDST1 activity by nAb13 may be caused by ordering of the D319−V332, C486−G512 and T528−L538 loops, which together form one face of the deacetylase cleft, and likely contact HS during catalytic processing (Supplementary Figs. 15, 17). Pre-organization of this interface by nAb13 would remove a substantial entropic penalty to HS engagement at the deacetylase domain, thereby enhancing the typically rate-limiting NDST1 deacetylation step[18]. Accordingly, Michaelis-Menten kinetics for NDST1-nAb13 with respect to K5 showed ~86% increased $k_{cat}$ compared to NDST1 alone (0.26 min$^{-1}$ vs 0.14 min$^{-1}$), indicating improved turnover efficiency. Conversely, NDST1-nAb7 displayed a ~3x reduced $k_{cat}/K_M$ compared to NDST1 alone (0.06 min$^{-1}$ mg$^{-1}$ mL vs 0.18 min$^{-1}$ mg$^{-1}$ mL), primarily driven by increased $K_M$, consistent with competitive inhibition of the deacetylase domain (Fig. 4F).

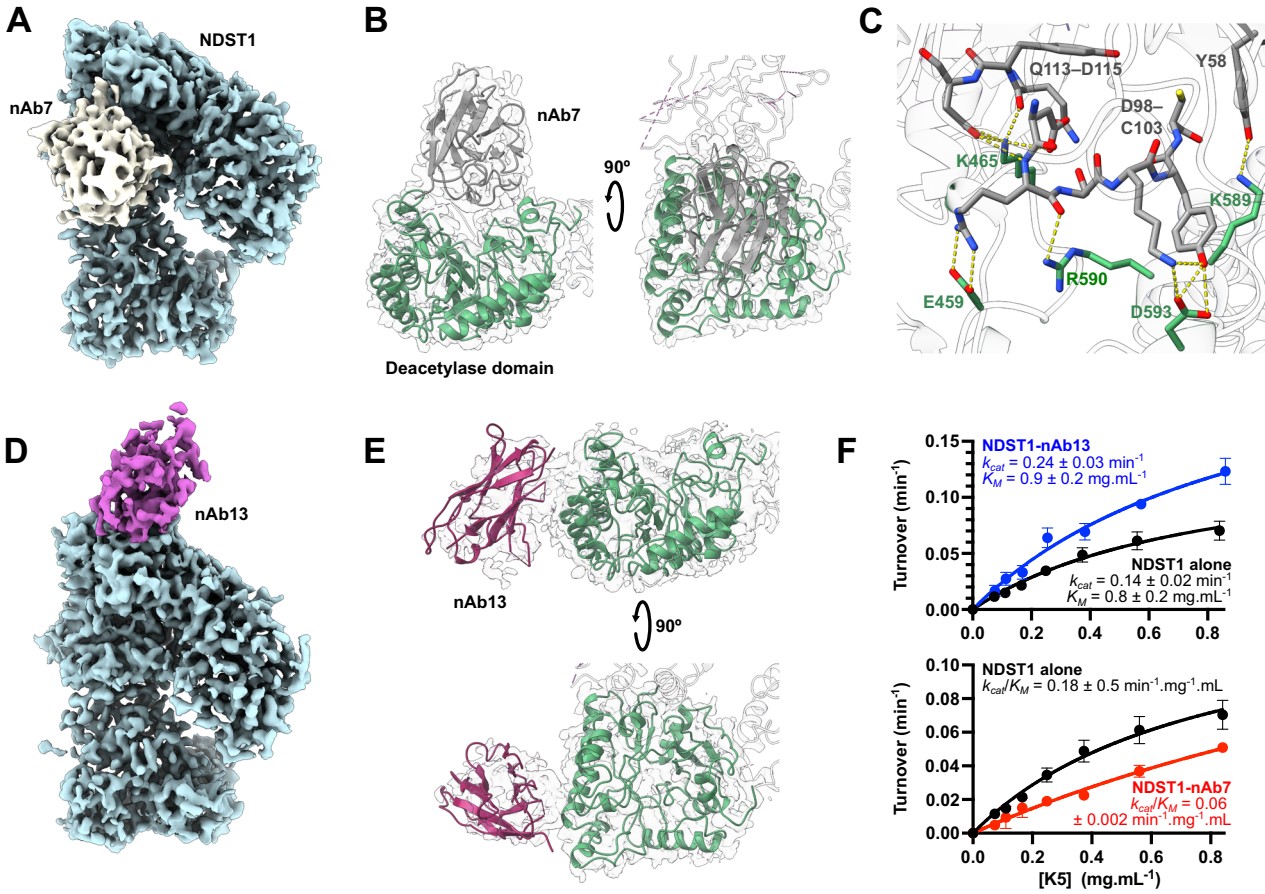

**Fig. 4 | Three-dimensional structures of NDST1 in complex with nAbs.**
**A** Coulombic density map of the NDST1-nAb7 complex contoured to 0.15 in ChimeraX. Density for NDST1 is colored blue for clarity. Density for nAb7 is colored gray. **B** Ribbon depiction of the NDST1-nAb7 complex, showing occlusion of the NDST1 deacetylase cleft by the bound nAb. **C** Molecular interactions at the NDST1-nAb7 interface. H-bonds depicted by yellow dashes. Density for this interface is shown in Supplementary Fig. 13a. **D** Coulombic density map of the NDST1-nAb13 complex contoured to 0.14. Density for nAb13 is colored purple. **E** Ribbon depiction of the NDST1-nAb13 complex, showing binding to the side of the NDST1 cleft. **F** Pseudo-first order Michaelis-Menten kinetics for NDST1, NDST1-nAb7 and NDST1-nAb13. nAb13 bound NDST1 shows enhanced turnover (higher $k_{cat}$) compared to NDST1 alone. nAb7 bound NDST1 shows poorer substrate engagement (higher $K_M$) compared to NDST1 alone. Datapoints are mean ± s.d. for $N$ = 4 technical replicates run on the same assay plate. Uncertainties for kinetic constants represent standard errors of curve fitting. Source data are provided within the Source Data file.

## NDST1-HS interactions analyzed by molecular docking

Despite extensive trials, we were unable to resolve an experimental NDST1-HS complex by cryo-EM. We thus employed molecular docking to gain insights into NDST1 interactions with HS substrates, using GlycoTorch Vina[41] to model the binding of an unsulfated octa-saccharide [GlcNAc-GlcA]$_4$ (hereafter NS0) or di-N-sulfated octa-saccharide [GlcNAc-GlcA]$_2$-[GlcNS-GlcA]$_2$ (NS2) to each catalytic domain of the nAb free NDST1 model.

Both NS0 and NS2 docked well into the NDST1 deacetylase cleft, placing their nonreducing termini towards the NTD, close to the channel formed at the deacetylase/NTD interface (Fig. 5A; Supplementary Fig. 11e). The resulting contact surface incorporated residues from both the NDST1 deacetylase and N-terminal domains (Supplementary Fig. 17a). Notably, the second GlcNAc (from reducing end) of each octasaccharide was found to dock with its N-acetyl moiety close to a catalytic metal center, in a position well-placed for deacetylation (Fig. 5B). Whilst waters were not explicitly considered during docking, this ligand pose supports a mechanism (based on Agd3 homology) in which a metal coordinated water is deprotonated by the nearby side-chain of base D319, inducing nucleophilic attack upon the GlcNAc N-acetyl center. Breakdown of the resulting oxyanion intermediate would be aided by the nearby acid H529, leading to the formation of GlcN with concomitant release of acetate. This mechanism is consistent with the inactivity of NDST1 deacetylase mutants (Fig. 3F), as well as previously proposed mechanisms for metal-dependent deacetylases[38,39].

Docking in the NDST1 sulfotransferase domain showed both NS0 and NS2 closely tracking the active site cleft (Fig. 5C; Supplementary Fig. 17b). The trajectory of both octasaccharides was similar to that observed for HS complexed with O-sulfotransferases HS3ST1 and HS3ST3 (PDB accessions 3UAN and 6XL8 respectively; Supplementary Fig. 17c), highlighting the conserved binding modes of these homologous enzymes. Although we did not observe close approach of docked NS0 and NS2 to the bound PAP ligand (Fig. 5D; ~9.5 Å between the PAP 5' phosphorous and the closest GlcNAc nitrogen), it is plausible that HS chains containing deacetylated GlcN would be able to approach closer to facilitate sulfate transfer from a sulfated PAPS donor.

## Sulfotransferase domain interactions initiate HS substrate engagement

Coordination between the deacetylase and sulfotransferase activities of NDST1 is essential for efficient HS modification and GlcNS cluster formation[14,16]. Unexpectedly, we found that the active sites of the NDST1 deacetylase and sulfotransferase domains project in opposing directions in the bifunctional enzyme, effectively hindering direct transfer of deacetylated HS intermediates from one domain to another (Fig. 6A; Supplementary Fig. 17d).

To determine possible mechanisms involved in domain cooperativity, we used BLI to measure the interaction of full-length NDST1 to surface immobilized K5 polysaccharide. The binding of NDST1 to K5 was well-modelled by a biphasic on/off model, with $K_D$s of 234 nM and 679 nM respectively (Fig. 6B; Table 2). Whilst biphasic interactions are

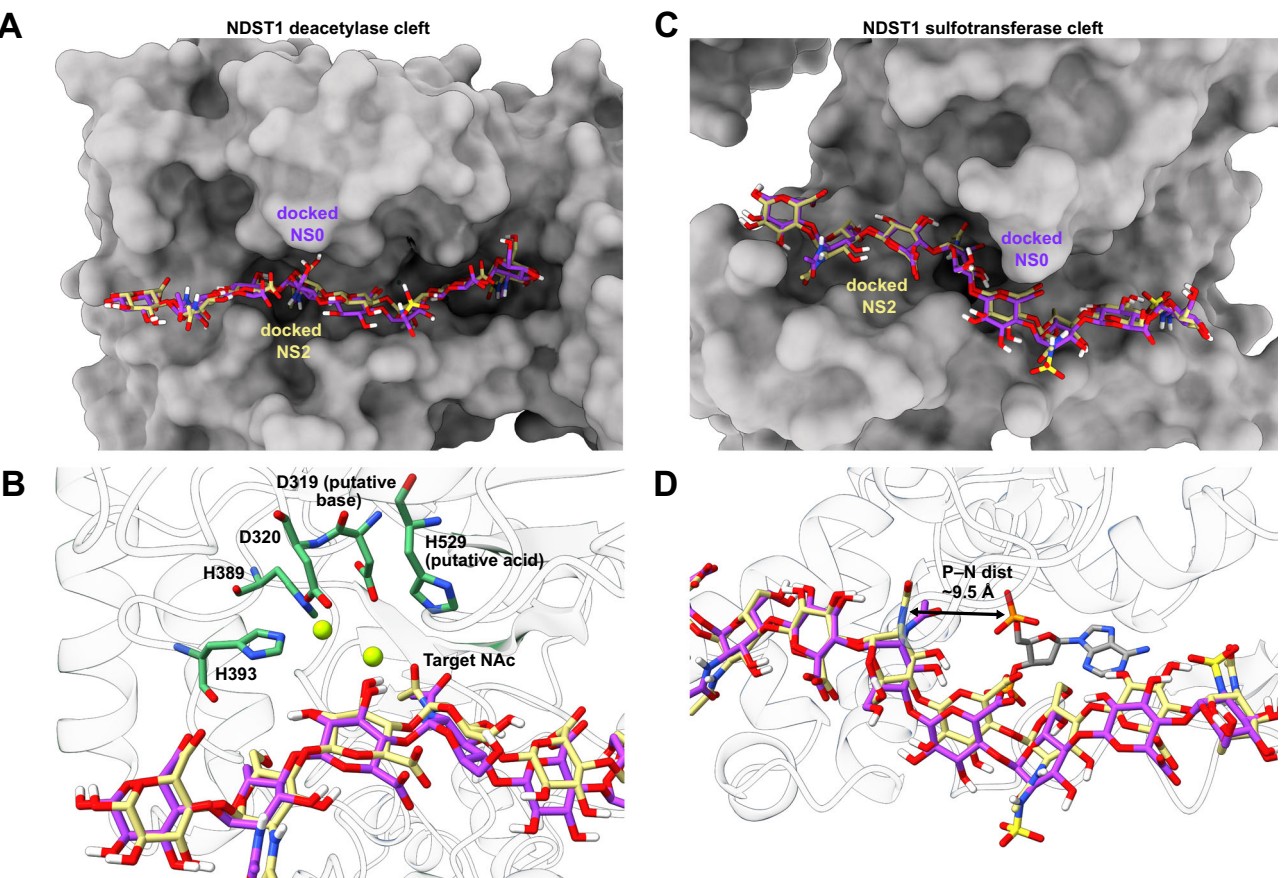

**Fig. 5 | Computational docking of octasaccharides in active site clefts of NDST1. A** Calculated docking poses for NS0 (purple) and NS2 (yellow) in the NDST1 deacetylase domain, showing similar carbohydrate trajectories within the binding cleft. **B** Close up of the deacetylase active site center, showing placement of the NAc substrate next to the catalytic amino acids and Ca$^{2+}$ ions. **C** Calculated docking poses for NS0 (purple) and NS2 (yellow) in the NDST1 sulfotransferase domain. **D** Close up of the sulfotransferase active site center. The docked carbohydrates remain ~9.5 Å distant from the bound PAP, possibly due to presence of GlcNAc, rather than GlcN on both NS0 and NS2.

not necessarily unusual for a multifunctional enzyme, we were intrigued by the substantial difference between the nanomolar $K_D$s measured for direct NDST1-K5 binding, compared to the high-micromolar $K_M$s determined for enzymatic NDST1-K5 processing (Fig. 1C). Although $K_D$ and $K_M$ are not directly analogous, both constants quantitate interactions between proteins and ligands. A substantial difference between these values for NDST1-K5 suggested the presence of additional binding events that might take place alongside enzymatic processing.

HS-protein interfaces are typically characterized by charge-based interactions between anionic polysaccharides and cationic surfaces on proteins. Nascent heparosan polysaccharides lack sulfates, but still possess considerable negative charge due to their repetitive carboxylate moieties. We thus assessed the binding of NDST1 to K5 polysaccharide in the presence of increasing concentrations of NaCl, to disrupt charge-based interactions. NaCl concentrations above ~300 mM essentially abrogated all binding, consistent with an electrostatic interface between NDST1 and HS, and in line with functional reports that NaCl strongly inhibits NDST1 catalysis[42,43] (Fig. 6C).

To identify regions on NDST1 that may bind HS via charge, we computed an electrostatic surface for NDST1 using the nAb-free structure, which revealed substantial cationic potential around the sulfotransferase cleft, contrasting with neutral to net-anionic potential at the deacetylase cleft (Fig. 6D). Whilst such charge-based considerations imply initial HS binding at the sulfotransferase domain, deacetylation is known to occur first during the NDST1 catalytic cycle (Supplementary Fig. 1c). We thus reassessed NDST1-K5 binding in the presence of 10 μM nAb7 or nAb13 (≫10x $K_D$ for both nAbs), which bind to different epitopes of the NDST1 deacetylase domain, with nAb7 also binding in a mode that occludes the polysaccharide substrate. As with full length NDST1, nanomolar K5 affinities were measured for both NDST1-nAb7 and NDST1-nAb13, thereby ruling out the NDST1

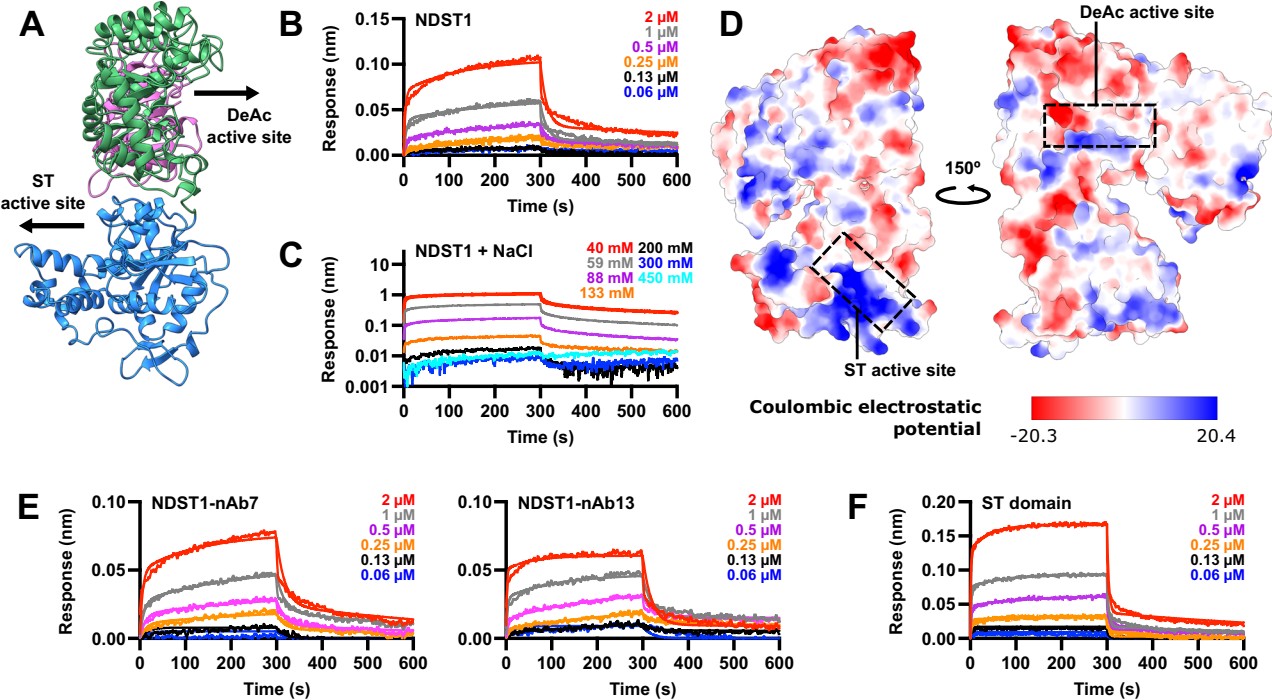

**Fig. 6 | NDST1 domain organization and substrate binding analysis. A** Ribbon diagram of the NDST1 structure, showing the opposing orientations of the deacetylase (DeAc) and sulfotransferase (ST) domain active sites. **B** BLI analysis of NDST1 binding to surface-immobilized K5 polysaccharide, showing biphasic on/off kinetics. **C** NDST1 binding to K5 is abrogated at NaCl concentrations above ~300 mM, suggesting a mainly electrostatic interaction. **D** Electrostatic surface potential of NDST1, calculated in ChimeraX. The sulfotransferase cleft contains a large positively charged region well-suited for binding HS. **E** NDST1-nAb7 and NDST1-nAb13 binding to surface-immobilized K5. Similar nanomolar binding affinity is observed for the two NDST1-nAb complexes compared to NDST1 alone. **F** Truncated NDST sulfotransferase domain binding to surface-immobilized K5. No substantial difference in affinity was observed compared to full-length NDST1. BLI binding constants are summarized in Table 2. Source data for BLI traces are provided within the Source Data file.

**Table 2 | NDST1 binding affinities for K5 with full length NDST1 protein, in the absence and presence of deacetylase domain binders nAb7 or nAb13, and truncated NDST1 sulfotransferase domain. Binding constants calculated from on/off rates from global curve fitting across 6 protein concentrations**

| Binder | 1st interaction (slow on/off) | | | 2nd interaction (fast on/off) | | |
|---|---|---|---|---|---|---|
| | $k_{assoc}$/M$^{-1}$ s$^{-1}$ | $k_{dis}$/s$^{-1}$ | $K_D$/nM | $k_{assoc}$/M$^{-1}$s$^{-1}$ | $k_{dis}$/s$^{-1}$ | $K_D$/nM |
| NDST1 alone | $3.5 \pm 0.1 \times 10^3$ | $8.3 \pm 0.1 \times 10^{-4}$ | $234 \pm 8$ | $1.1 \pm 0.02 \times 10^5$ | $7.3 \pm 0.1 \times 10^{-2}$ | $679 \pm 17$ |
| NDST1-nAb7 | $2.9 \pm 0.2 \times 10^3$ | $1.9 \pm 0.02 \times 10^{-3}$ | $673 \pm 39$ | $6.1 \pm 0.2 \times 10^4$ | $5.6 \pm 0.1 \times 10^{-2}$ | $921 \pm 34$ |
| NDST1-nAb13 | $8.5 \pm 0.3 \times 10^3$ | $3.2 \pm 0.1 \times 10^{-4}$ | $37 \pm 2$ | $2.7 \pm 0.1 \times 10^5$ | $5.1 \pm 0.05 \times 10^{-2}$ | $191 \pm 6$ |
| ST domain | $1.2 \pm 0.02 \times 10^4$ | $1.6 \pm 0.01 \times 10^{-3}$ | $137 \pm 3$ | $3.5 \pm 0.1 \times 10^5$ | $2.6 \pm 0.03 \times 10^{-1}$ | $745 \pm 23$ |

Uncertainties represent standard errors of curve fitting.

deacetylase cleft as a site of initial substrate binding (Fig. 6E; Table 2). Finally, we tested the binding of truncated NDST1 sulfotransferase domain alone to K5 polysaccharide. The NDST1 sulfotransferase domain displayed essentially equivalent nanomolar affinity for K5 as compared to full-length enzyme, further reinforcing the primacy of this domain in driving NDST1 polysaccharide interactions (Fig. 6F; Table 2).

Taken together, our data indicate that initial, high-affinity NDST1-HS binding likely takes place at the sulfotransferase domain, despite deacetylation being the first step of the enzymatic processing cycle.

## Discussion

Mammalian HS biosynthesis is a complex multistep process, which involves the coordinated activity of multiple enzymes along the ER-Golgi pathway. NDSTs are key enzymes in the early stages of HS maturation, which convert GlcNAc sugars in nascent heparosan to GlcNS. Clusters of GlcNS laid down by NDSTs play a major role in directing further downstream HS processing enzymes, thus understanding NDST function fundamentally informs upon how cells create the complex and heterogenous HS sequences required by biology.

We have carried out a detailed structural and functional characterization of NDST1, the dominant human NDST isoform, revealing a 3-domain architecture consisting of a non-catalytic NTD, in addition to catalytic deacetylase and sulfotransferase domains. The previously unsolved deacetylase domain of NDST1 is characterized by a His-His-Asp triad typical of metal-dependent sugar deacetylases. Herein, we have assigned the presence of catalytic $Ca^{2+}$ coordinated by this triad, based on its high affinity carry through from protein purification. However, NDST1 has also been shown to be active in the presence of other divalent ions, including $Mn^{2+}$, $Mg^{2+}$ and $Co^{2+}$[18]. It is possible that native NDST1 within the Golgi apparatus, which is enriched in many divalent cations[44], may employ a variety of metal ions to facilitate deacetylation.

Computational docking with HS octasaccharides suggests that the hitherto undescribed NDST1 NTD may play a role in polysaccharide engagement, by contributing to the formation of a substrate binding cleft adjacent to the deacetylase domain. The presence of carbohydrate binding module (CBM) domains that assist substrate binding is common amongst polysaccharide processing enzymes[45]. Indeed, the fungal deacetylase Agd3, which is homologous to the NDST1 deacetylase domain, possesses an N-terminal CBM that enhances substrate attachment[38]. Whilst this N-terminal domain of Agd3 extends a linear substrate cleft projecting from its catalytic site, the cleft formed by the analogous NDST1 NTD is kinked with respect to the deacetylase domain (Supplementary Fig. 11e), potentially reducing its effectiveness in aiding HS binding. Accordingly, we find that initial NDST1 engagement of HS is driven by interactions at its sulfotransferase, rather than deacetylase domain, despite the latter operating first in the catalytic cycle. A further role of the NDST1 NTD may be in enzyme oligomerization. Vallet et al. have recently reported the intriguing observation of stable NDST1 homodimers[46], which interact via a C2 symmetric interface involving the NTD and sulfotransferase domains (Supplementary Fig. 18). NDST1 oligomerization may be mechanistically significant during catalytic turnover (see below), thus implicating the NTD in functions beyond simple substrate binding.

A distinctive feature of the NDST1 protein architecture is the opposing orientation of its catalytic deacetylase and sulfotransferase domains, which effectively precludes a simple substrate transfer mechanism for bifunctionality. Our data indicate that initial NDST1-HS binding occurs at the sulfotransferase domain via an electrostatic interface, which occurs with higher affinity than that required for enzymatic conversion. This high-affinity interaction may help NDST1 enzymes accumulate on heparosan polysaccharides prior to processing. Initial sulfotransferase binding also provides a ready mechanism for bifunctionality, which obviates the need for direct domain-to-

domain substrate transfer. Putatively, deacetylation of HS polysaccharides anchored to the NDST1 sulfotransferase domain would produce GlcN intermediates that remain bound to the enzyme. Such intermediates, already located at the NDST1 sulfotransferase domain, would consequently be ideally positioned for immediate sulfation to produce GlcNS. Since initial NDST1-HS interactions are predominantly electrostatic, they may also bind newly formed GlcNS preferentially, further raising the intriguing possibility of a molecular ratcheting mechanism that can lead to the formation of GlcNS clusters (Supplementary Fig. 19). Consistent with this, we note that NDST1 processing of HS, which contains some N-sulfation, shows improved $K_M$ and $k_{cat}$ compared to its processing of K5, which entirely lacks N-sulfation (Fig. 1C).

The precise nature of HS coordination between the NDST1 deacetylase and sulfotransferase domains remains to be elucidated. In the absence of direct visualization of NDST1-HS binding, two models may be considered for how a polysaccharide chain can access both NDST1 catalytic sites in a coordinated fashion. A 1:1 model, in which a single HS chain loops around NDST1 to access both catalytic domains, appears somewhat implausible, given the entropic penalty that would arise from constraining extended HS polysaccharides (Supplementary Fig. 17, 19). Alternatively, a 2:1 (or >2:1) model, in which the catalytic cycle is shared between multiple NDST1 enzymes, might operate. Such a model accords well with recent observations of catalytically competent NDST1 homodimers by Vallet et al., in which an alternative arrangement of NDST1 catalytic domains is created by the dimeric interface, although the active sites remain some 60 Å apart (Supplementary Fig. 18)[46]. Formation of NDST1 oligomers may provide a route to direct HS chains, initially anchored to the sulfotransferase domain of one protomer, to the deacetylase domain of a neighbor, thereby enabling functional coupling in a manner distinct from the 1:1 model (Supplementary Fig. 19). Whilst stable NDST1 oligomers were not observed in this work, possibly due to differences in construct design (6xHis-tagged vs phosphatase-tagged) or expression system (insect vs mammalian) compared to Vallet et al., transient oligomerization may still operate in our system to coordinate NDST1 activities during catalysis. Interestingly, low (~7.3) $\mu M$ $K_M$s were measured by Vallet et al. for processing of heparosan by partially dimeric NDST1, contrasting with the high $\mu M$ values recorded here (Fig. 1C), and consistent with improved functionality for a dimer (albeit caution must be exercised when comparing values measured using different assays). We also do not discount the possibility of further oligomeric NDST1 states within intracellular milieu, formed by the preorganization of enzymes on Golgi membranes. Detailed analysis of NDST1-HS interactions across multiple polysaccharide sequences and NDST1 oligomerization states will be required to dissect the consequences of different substrate-binding modes and stoichiometries on catalysis.

One final aspect of NDST1 function that has not been addressed here is the formation of free GlcN residues, which typically comprise ~0.7–4% of HS sequences[47]. These rare GlcN residues may arise from NDST1-mediated deacetylation of heparosan that occurs without initial sulfotransferase domain binding. Alternatively, stalling of the NDST1 catalytic cycle at the sulfotransferase step (e.g. due to lack of PAPS), may also cause premature release of deacetylated intermediate. Both possibilities are likely to be rare, given the weaker HS binding capacity of NDST1 deacetylase domains (Fig. 1C), and the abundance of sulfation reactions within cells requiring a stable pool of PAPS, respectively. Such mechanistic rarity is consistent with the paucity of GlcN in mature HS.

Important to our characterization of NDST1 was the de novo generation of a panel of nAb binders, which aided our analysis of NDST1-HS interactions, and shed light on allosteric factors that influence NDST1 catalytic activity. Beyond their clear utility for functional analysis, these nAbs provide a potential toolkit for functional studies of NDST1 biology e.g. as probes to track enzymes in situ[48]. Notably, we

have isolated nAbs that both inhibit (nAb7), or enhance (nAb13) NDST1 activity. Given the paucity of high-quality tools to perturbate HS biosynthesis in cells, nAbs may represent useful, genetically encodable modules to up- or downregulate HS modifications in a reversible manner. In situ use may require higher binding affinities than the high-nM potencies currently displayed by nAb7 and nAb13. Thus affinity maturation steps will likely form part of future efforts to develop HS modulating nAb binders[49].

With our characterization of bifunctional NDST1, all enzyme families involved in the mammalian HS biosynthesis pathway (linker biosynthesis, polymerization, N-sulfation, epimerization, 2O-, 3O- and 6O-sulfation) now have at least one experimentally resolved representative[14,50–56]. The GAGosome hypothesis, first proposed over 20 years ago, posits the existence of a large macromolecular assembly of biosynthesis enzymes tasked with constructing HS in a coordinated fashion[17]. Although some HS biosynthesis enzyme interactions have been inferred by coimmunoprecipitation or immunofluorescence, the only multimers to have been directly visualized remain those between the obligate dimeric EXT enzymes[14–16]. Our observation of initial high-affinity binding interactions between NDST1 and HS hints at an alternative model for the GAGosome, in which HS itself may induce the formation of complexes that transiently assemble around newly synthesized polysaccharide chains. Detailed analysis of individual HS biosynthesis enzymes such as NDST1 provides a vital basis for understanding mechanisms of cooperativity within complex multi-component systems such as the GAGosome, and will underpin future studies of HS biosynthesis within in situ cellular contexts.

## Methods

### Cloning and gene expression—NDST1

The pDONR221-NDST1 plasmid was obtained from the DNASU repository (Arizona State University), from which a cDNA fragment encoding solubilized NDST1 (residues 79–882) was amplified using Q5 high fidelity DNA polymerase (New England Biolabs). The NDST1(79–882) fragment was subcloned using Infusion (Takara Bio) into the pOMNIBac transfer vector (Geneva Biotech), behind a honeybee melittin secretion peptide, 6xHis tag and TEV cleavage site. NDST1 mutants were derived from the wild-type NDST1(79–882) construct by site-directed mutagenesis using a PCR-based method[57]. C-terminal Avitagged NDST1 was generated by PCR with tailed primers, followed by subcloning into pOMNIBac. All constructs were verified by Sanger sequencing before further use. DNA and protein sequences for NDST1 constructs are available in Supplementary Table 2. Primers used for cloning, subcloning and mutagenesis are outlined in Supplementary Table 3.

Recombinant bacmids were produced via the Tn7 transposition method in DH10EMBacY cells (Geneva Biotech), and purified using the PureLink miniprep kit (Invitrogen) according to manufacturers' protocols. V1 baculovirus stocks were produced by transfection of bacmid into low-passage ExpiSf9 cells (Invitrogen; *Spodoptera frugiperda)* in adherent format using FuGENE HD (Promega) at a ratio of ~2 mg bacmid DNA to ~5 μL FuGENE. V1 to V2 virus amplification was carried out in ExpiSf9 cells in suspension culture, at a density of ~1–2 × 10^6 mL^−1. Viral propagation was tracked using the YFP marker encoded by the EMBacY virus. Optimum baculovirus amplification was typically achieved 72 h after inoculation, when cells appeared 60 + % fluorescent. V2 virus stocks were harvested by centrifugation at 200g for 5 min at 4 °C to remove cells, and supplemented with 2% v/v FBS for storage.

Large scale gene expressions were carried out in High-Five (Invitrogen; *Trichoplusia ni*) or ExpiSf9 cells in log-phase growth. V2 baculovirus was added to cells at a multiplicity of infection > 1, and the resulting infection tracked using the EMBacY YFP marker. Cultures were typically harvested at ~72 h, or when cells showed ~80% or greater fluorescence.

### Protein purification—NDST1

Solubilized recombinant NDST1 was purified from conditioned insect cell media. Typically, ~2–3 L of insect cell culture was cleared of cells by centrifugation at 200g for 15 min at 4 °C, followed by further clearing of debris by centrifugation at 5000g for 60 min at 4 °C. Clarified media was supplemented with DL-dithiothreitol (DTT; 1 mM) and phenylmethylsulfonyl fluoride (PMSF; 0.2 mM), before loading onto a 5 mL Histrap Excel column (Cytiva), pre-equilibrated with Histrap buffer A (20 mM Tris pH 8.0, 500 mM NaCl, 20 mM Imidazole, 1 mM DTT), at a rate of 5 mL min^−1. The loaded Histrap column was washed with 10 column volumes (CV) Histrap buffer A, before eluting with a linear gradient of Histrap buffer A to Histrap buffer B (20 mM Tris pH 8.0, 500 mM NaCl, 500 mM Imidazole, 1 mM DTT) over 10 CV. NDST1 containing fractions, as determined by SDS-PAGE, were pooled, diluted 10-fold in milli-Q water, and adjusted to pH 6.5 by the addition of 1 M MES acid, before loading onto a Hitrap SP XL column (Cytiva) pre-equilibrated with ion exchange (IEX) buffer A (20 mM MES pH 6.5, 1 mM DTT). Loaded Hitrap SP XL columns were washed with 10 column volumes (CV) IEX buffer A, before eluting with a linear gradient of IEX buffer A to IEX buffer B (20 mM MES pH 6.5, 1500 mM NaCl, 1 mM DTT) over 10 CV. NDST1-containing fractions were pooled again and digested overnight at room temperature with TEV protease at 1:100 mass ratio TEV:NDST1 to cleave the 6xHis tag. TEV digested NDST1 was rerun over a 5 mL Histrap Excel column pre-equilibrated with Histrap buffer A, which was then washed with 4 CV Histrap buffer A. Flowthrough and wash fractions were pooled and concentrated to 2 mL volume using a 30 kDa molecular weight cut-off (MWCO) Vivaspin concentrator (Cytiva). Final purification was carried out by size-exclusion chromatography (SEC), using a HiLoad 16/600 Superdex 200 pg column (Cytiva), pre-equilibrated and run in SEC buffer (20 mM MES pH 6.5, 200 mM NaCl, 1 mM DTT). Pure NDST1 containing fractions were pooled, adjusted to <100 mM NaCl using IEX buffer A, and concentrated to ~10 mg/mL using a 30 kDa MWCO Vivaspin concentrator. Small aliquots were flash frozen using liquid nitrogen, and stored at −80 °C for further use.

Mutant NDST1 constructs were purified in the same way as for wild type protein. Biotinylated NDST1 was prepared using the C-terminal Avitagged protein, which was treated with the BirA biotin-protein ligase reaction kit (Avidity, LLC) following the manufacturers protocol. Biotinylated NDST1 was purified by a final round of SEC following the BirA reaction.

### Cloning and gene expression – Sult1a1

A plasmid encoding cDNA for *Sult1a1* from *Rattus norvegicus* was obtained from GenScript Biotech. A cDNA fragment encoding for wild type *Sult1a1* was subcloned into the pET15b backbone, behind a thrombin cleavage site, using Ndel and BamHI restriction sites. Mutations K65E and R68G, which prevent the formation of an inhibitory Sult1a1-PAP-4-methylumbelliferone complex[58], were introduced into the plasmid by site-directed mutagenesis. Primers used for cloning and mutagenesis are outlined in Supplementary Table 3. The final construct was verified by Sanger sequencing before further use.

For expression, *Sult1a1*-K65E-R68G was used to transform BL21 Gold (DE3) chemically competent cells (Agilent) by heat shock. Transformed cells were grown in TB media containing 100 μg mL^−1 ampicillin at 37 °C with shaking, until reaching an OD_{600} of 0.8–1.0. Cultures were induced by the addition of isopropyl-b-thiogalactoside (IPTG; Generon) to a final concentration of 0.5 mM, then grown overnight at 16 °C with shaking. Cultures were harvested by centrifugation at 5000g for 15 min at 4 °C, and pellets stored at −80 °C until further processing.

### Protein purification—SULT1A1-K65E-R68G

Sult1a1-K65E-R68G pellets were thawed and resuspended in 100 mM Tris pH 7.6, 1 mM DTT containing 120 kU DNase I (Sigma) and

cOmplete™ EDTA-free Protease Inhibitor Cocktail (Roche). Resuspended cell pellets were lysed via two rounds of cell disruption at 30 kpsi (Constant Systems) at 4 °C. Cell debris was removed by centrifugation at 14,000$g$ at 4 °C for 30 min, and supernatant filtered prior to loading onto a 5 mL Histrap FF column, pre-equilibrated in Histrap buffer A (100 mM Tris pH 7.6, 300 mM NaCl, 20 mM imidazole, 1 mM DTT). The loaded column was washed with 5 CV of buffer A prior to elution of Sult1a1-K65E-R68G using a linear gradient of 0–100% Histrap buffer B (100 mM Tris pH 7.6, 300 mM NaCl, 250 mM imidazole, 1 mM DTT) over 10 CV. Sult1a1-K65E-R68G containing fractions were pooled and digested overnight at room temperature with thrombin at 1:100 mass ratio thrombin:Sult1a1-K65E-R68G. Thrombin-cleaved Sult1a1-K65E-R68G was passed over a 5 mL Histrap FF column pre-equilibrated in buffer A, and the column was washed with 5 CV of the same buffer. The flow-through and wash fractions, containing de-tagged Sult1a1-K65E-R68G, were collected and pooled before being concentrated using a 10 kDa MWCO spin filter. Sult1a1-K65E-R68G was further purified by SEC using a HiLoad 16/600 Superdex 75 pg column pre-equilibrated in SEC buffer (100 mM Tris pH 7.6, 1 mM DTT). Fractions containing Sult1a1-K65E-R68G were assessed by SDS-PAGE and Coomassie staining before being pooled and concentrated to 6 mg mL$^{-1}$ using a 10 kDa MWCO spin filter. Aliquots of concentrated protein were flash frozen in LN$_2$ and stored at −80 °C until further use.

## Expression—heparin lyase II

The gene encoding heparin lyase II from *Pedobacterium heparinus*, inserted into pRSET-A (gifted from Marcelo Lima, University of Keele), was transformed into chemically competent C41(DE3) pLysS cells by heat shock. 1 L cultures of TB media, containing 100 µg mL$^{-1}$ ampicillin and 50 µg mL$^{-1}$ chloramphenicol were inoculated and grown at 37 °C, 250 rpm with shaking until reaching an OD$_{600}$ of 0.6–0.8, whereupon they were induced with 0.5 mM IPTG before further incubation at 22 °C for 16 h, 250 rpm with shaking. Cultures were harvested at 6000 $g$ for 15 min at 4 °C. Pellets were stored at −80 °C until further use.

## Protein purification—heparin lyase II

Thawed cell pellets were resuspended in 20 mM sodium phosphate pH 7.9, 500 mM NaCl, 5 mM imidazole supplemented with DNAse I (Sigma) and cOmplete™ EDTA-free protease inhibitor (Roche). Cells were lysed via two rounds of cell disruption at 30 kpsi, then centrifuged at 14,000$g$, 4 °C, for 30 min to remove cell debris. The supernatant was filtered and loaded onto a 5 mL HiTrap FF column pre-equilibrated in Histrap buffer A (20 mM sodium phosphate pH 7.9, 500 mM NaCl, 5 mM imidazole), which was washed with 5 CV of the same buffer prior to elution with a linear gradient of 0–100% Histrap buffer B (20 mM sodium phosphate pH 7.9, 500 mM NaCl, 250 mM imidazole) over 10 CV. Fractions containing heparinase lyase II were pooled, diluted with 10 volumes of IEX buffer A (20 mM MES pH 6.0) and then loaded onto a 5 mL Hitrap SP XL column (Cytiva) pre-equilibrated in the same buffer. Elution was conducted with a linear gradient of 0–100% IEX buffer B (20 mM MES pH 6.0, 1.5 M NaCl) over 20 CV. Fractions containing heparin lyase II were pooled and concentrated using a 30 KDa MWCO spin filter, then applied to a HiLoad 16/600 Superdex 200 pg column equilibrated and run with SEC buffer (20 mM sodium phosphate pH 6.8, 150 mM NaCl). Heparin lyase II containing fractions were confirmed by SDS-PAGE, before being pooled and concentrated to ~2 mg/mL using a 30 kDa MWCO spin filter. Purified protein was stored at 4 °C until required.

## Circular dichroism

The CD spectrum of 250 µg mL$^{-1}$ NDST1, and mutants thereof, were acquired in 20 mM phosphate buffer pH 7.4 using a Chirascan V100 (Applied Photophysics), equipped with a 0.2-mm path length quartz cuvette (Hellma, United States). All spectra were recorded with a scanning speed of 1 nm s$^{-1}$ with 0.5 nm resolution, between 190 and 260 nm using Chirascan control software version 2.02. Spectra are presented as the mean of five independent scans smoothed with second-order polynomial smoothing through 11 neighbors, using GraphPad Prism 10.1.0.

## Inductively coupled plasma optical emission spectrometry (ICP-OES)

150 µg mL$^{-1}$ NDST1 was digested by the addition of 0.25 volumes of HNO$_3$ in a total volume of 4 mL. Transferrin or apotransferrin (440 µg mL$^{-1}$) controls were digested with the addition of 0.25 volumes of HNO$_3$ and diluted two-fold across 5 concentrations in 1.25% HNO$_3$. Following digestion, each sample was filtered using a 0.25 mm PVDF filter prior to measurement of Ca ($\lambda_{em}$ = 422.673 nm), Cu ($\lambda_{em}$ = 324.754 nm), Fe ($\lambda_{em}$ = 260.709 nm), Mg ($\lambda_{em}$ = 279.553 nm), Mn ($\lambda_{em}$ = 279.482 nm) and Zn ($\lambda_{em}$ = 206.2 nm) using an Agilent Technologies 700 Series ICP-OES. ARISTAR® multi-element quality control standard (VWR), containing metal ions at a concentration of 1 mg mL$^{-1}$ each, was serially diluted 2-fold across 5 concentrations covering the range of 0.0625 − 1 ppm to allow the construction of a calibration curve for each element. Suitable wavelengths were chosen where interferences were not observed, and a sufficient signal:noise ratio was observed above baseline. All samples were obtained as a mean of 3 replicates under a plasma argon flow rate of 15 L min$^{-1}$, auxiliary argon flow rate of 1.5 L min$^{-1}$ and a nebulizer argon flow rate of 0.75 L min$^{-1}$.

## Purification of K5 polysaccharide

K5 polysaccharide was purified from *E. coli* strain Bi 8337/41(O10:K5:H4). Typically, 1 L of LB media was inoculated with 10 mL of *E. coli* strain Bi 8337/41(O10:K5:H4), grown for 16 h at 37 °C with shaking, before being incubated for a further 55 h at 37 °C. Cells were removed via centrifugation at 8000$g$ for 15 min at 4 °C and the supernatant collected. The supernatant was filtered and diluted with 1 volume of IEX buffer A (20 mM sodium acetate pH 4.0, 50 mM NaCl), further adjusted to pH 4.0 with acetic acid as required, before being loaded onto a C10/20 column (Cytiva) packed with DEAE Sepharose resin (Biorad), pre-equilibrated with the same buffer. Following loading, the column was washed with IEX buffer A and bound K5 polysaccharide eluted in a single step with IEX buffer B (20 mM sodium acetate pH 4.0, 1 M NaCl). The eluant was mixed with 3 volumes of ice-cold absolute ethanol and stored overnight at −20 °C. The resulting precipitate containing K5 polysaccharide was collected by centrifugation at 13,000$g$, 4 °C for 30 min and resuspended in dH$_2$O before being lyophilized. Lyophilized K5 was dissolved in 1 M NaCl to 15 mg mL$^{-1}$ and pH adjusted to 9.5 with 1 M NaOH. 30% hydrogen peroxide was added to a final concentration of 1.5%, before incubation overnight at room temperature. Bleached K5 polysaccharide was recovered by the addition of 3 volumes of ice-cold ethanol followed by storage at −20 °C overnight. The precipitate was collected by centrifugation at 13,000$g$ at 4 °C for 30 min and dialyzed against dH$_2$O, exchanged a minimum of three times using a 5 kDa MWCO dialysis membrane (Biodesign). Dialyzed K5 polysaccharide was filtered and lyophilized before being stored at −20 °C for further use. The purity of K5 was assessed using nuclear magnetic resonance (NMR). Polysaccharide was exchanged into D$_2$O (700 µL) and one-dimensional ($^1$H) spectra were recorded using a Bruker Avance NEO 600 MHz spectrometer fitted with a BBOH&F cryoprobe at 298 K.

## Fluorometric assay for determination of Sult1a1-K65E-R68G kinetics

An 8-point 2-fold serial dilution of PAP from 250 µM was incubated with 350 nM Sult1a1-K65E-R68G, 4 mM 4-methylumbellferyl (4MU)-sulfate in 50 mM Tris-HCl pH 7.5, 15 mM MgCl$_2$, 1 mM DTT, in black half area 96 well plates (Greiner) in a total volume of 20 µL. Controls omitting PAP were included and subtracted from each reaction prior to further analysis. Reactions were initiated by the addition of Sult1a1-K65E-R68G

and the change in fluorescence recorded using a Clariostar plate reader (BMG Labtech; software version 5.70 R2) using the 4MU fluorescence preset (Ex. 360 [b.p. 20] nm; Em. 450 [b.p. 30] nm). An 8 point 2-fold serial dilution standard curve of 4MU (from 10 μM) in 50 mM Tris-HCl pH 7.5, 15 mM MgCl$_2$ was used to calculate the evolution of 4MU during the reaction course. Initial reaction velocities at each substrate concentration were fitted to the Michaelis-Menten equation to derive $K_M$ and $V_{max}$. $k_{cat}$ was derived from $V_{max}$ using the relationship $V_{max}=k_{cat}*[E_t]$. All graphs and curve fittings were processed using GraphPad Prism 10.1.0.

## Fluorometric coupled enzyme activity assay for NDST1 and mutants

For the determination of pseudo-first order NDST1 kinetics with respect to polysaccharide substrate, varying concentrations of K5 or HS polysaccharide (low sulfated fraction; Iduron, GAG HS-I) were incubated with 250 nM NDST1, 350 nM Sult1a1-K65E-R68G, 20 μM PAPS and 4 mM 4MU-sulfate in 50 mM Tris-HCl pH 7.5, 15 mM MgCl$_2$, 1 mM DTT. Controls omitting polysaccharide were subtracted from reactions prior to further analysis. For determination of pseudo-first order kinetics with respect to PAPS, a 2-fold serial dilution of PAPS from 50 μM was incubated with 0.5 mg mL$^{-1}$ HS, 250 nM NDST1, 350 nM Sult1a1-K65E-R68G and 4 mM 4MU-sulfate in 50 mM Tris-HCl pH 7.5, 15 mM MgCl$_2$, 1 mM DTT. Controls omitting PAPS were subtracted from reactions prior to further analysis.

Evaluation of NDST1 mutants was performed using 1 μM enzyme in the presence of 350 nM Sult1a1-K65E-R68G, 20 μM PAPS, 0.5 mg mL$^{-1}$ K5 and 4 mM 4MU-sulfate in 50 mM Tris-HCl pH 7.5, 15 mM MgCl$_2$, 1 mM DTT. For evaluation of NDST1 activity with nanobodies, reactions were performed with an 8 point, 3-fold dilution of each nanobody from 90 μM in the presence of 250 nM NDST1, 350 nM Sult1a1-K65E-R68G, 20 μM PAPS, 0.5 mg mL$^{-1}$ K5 and 4 mM 4MU-sulfate in 50 mM Tris-HCl pH 7.5, 15 mM MgCl$_2$, 1 mM DTT.

The effect of nAb7 or nAb13 on NDST1 kinetics was evaluated by the addition of 30 μM nanobody to pseudo-first order NDST1 kinetics reactions with respect to K5 (see above). Reactions were incubated with 250 nM NDST1, 350 nM Sult1a1-K65E-R68G, 20 μM PAPS and 4 mM 4MU-sulfate in 50 mM Tris-HCl pH 7.5, 15 mM MgCl$_2$, 1 mM DTT. Controls omitting polysaccharide were subtracted from reactions prior to further analysis.

The effect of different metal ions on NDST1 activity was evaluated in the presence of 1 μM NDST1, 350 nM Sult1a1-K65E-R68G, 20 μM PAPS, 0.5 mg mL$^{-1}$ K5 and 4 mM 4MU-sulfate in 50 mM Tris, pH 7.4, where the reaction buffer had been supplemented with 50 mM of MgCl$_2$, CaCl$_2$ or H$_2$O (metal free control), or 20 mM EDTA or dipicolinic acid.

In all cases reactions were performed in black half-area 96 well plates (Greiner) with a total reaction volume of 20 μL. All reactions were initiated by the addition of polysaccharide before recording the change in fluorescence with a Clariostar plate reader (BMG Labtech; software version 5.70 R2) using the 4MU preset. An 8 point 2-fold serial dilution standard curve of 4MU (from 10 μM) in 50 mM Tris-HCl pH 7, 15 mM MgCl$_2$, was used to calculate the evolution of 4MU during the reaction timecourse.

For most datasets, initial reaction velocities at each substrate concentration were fitted to the Michaelis-Menten equation to derive $K_M$ and $V_{max}$ values. $k_{cat}$ was derived from $V_{max}$ using the relationship $V_{max}=k_{cat}*[E_t]$. For kinetics with the NDST1-nAb7 complex (Fig. 4F) where reaction velocities did not plateau (i.e. $[S] \ll K_M$), a combined $k_{cat}/K_M$ value was derived from the slope of a linear fit to the data. $K_i$ values for NDST1 with nAb5 and nAb7 were calculated using the Morrison quadratic equation, which applies when $K_I$ and $[E_t]$ are in similar ranges: $Y = V_O*(1-((((E_t+X+Q)-(((E_t+X+Q)^2)-4*E_t*X)^0.5))/(2*E_t)))$; $Q = (K_i*(1+([S]/K_M)))$. Graphs were processed and visualized using GraphPad Prism 10.1.0.

## Δ-Disaccharide analysis of time course-NDST1 treated K5

600 μg of K5 was incubated with, 250 μM PAPS and 250 nM NDST1 in 50 mM Tris-HCl pH 7.4, 15 mM MgCl$_2$ for 18 h at 37 °C with shaking. Aliquots of the digest reaction were taken at various timepoints and heat inactivated at 95 °C for 10 min, before being stored at −20 °C until analysis.

Heparin lyase II (2 ug) was added to each sample and incubated at 30 °C for a total of 24 h, with the addition a further 2 ug of Heparin lyase II after 8 h. Samples were heat denatured at 95 °C for 5 min and stored at −20 °C before analysis. Chromatographic separation of Heparin lyase II digested samples was performed using high performance anion exchange chromatography (HPAEC). Samples were made up to 1 mL in HPLC-grade H$_2$O (Fisher) prior to being injected onto a ProPac PA-1 analytical column (4×250 mm), pre-equilibrated in HPLC-grade H$_2$O, at a flow rate of 1 mL min$^{-1}$. The column was held under isocratic flow for 10 minutes, followed by elution of Δ-disaccharides using a linear gradient of NaCl from 0 to 2 M NaCl in HPLC-grade H$_2$O over 60 minutes. Elution was monitored by in-line UV detection of A$_{232}$ via the C = C unsaturated bond introduced by Heparin lyase digestion. Retention times were compared to Δ-disaccharide reference standards (Iduron). The column was washed extensively with 2 M NaCl and HPLC-grade H$_2$O in between runs.

## Biotinylation of K5 polysaccharide

K5 polysaccharide (4 mM) was biotinylated at the reducing end through a reaction with 1:1 molar ratio of N-(aminooxyacetyl)-N'-(D-Biotinoyl) hydrazine in the presence of 100 mM aniline, in 100 mM NaOAc pH 4.6[59,60]. The reaction was incubated for 48 h at 37 °C before free biotin was removed by desalting on a C 10/40 column packed with Sephadex G25 (Cytiva), pre-equilibrated in dH$_2$O. Separation was conducted over 1.2 CV with inline UV monitoring at 232 and 210 nm. Fractions eluting in the void volume corresponding to K5 were pooled and lyophilized. Biotinylation was assessed by dot blot; 2 μL of 1 mg mL$^{-1}$ K5 or biotinylated K5 was spotted onto a nitrocellulose membrane and allowed to dry, before being blocked with 5% (w/v) BSA in PBS-T (PBS, 0.05% Tween-20) for 1 h. The blocked nitrocellulose membrane was washed 3x with PBS-T before being incubated with 1:2000 HRP-streptavidin (Thermo 21124) in 1% (w/v) BSA in PBS-T for 30 min. Following three washes with PBS-T, the presence of biotin was detected by the addition of SuperSignal West Femto chemiluminescence substrate (Thermo), and imaged using a ChemiDoc system (BioRad).

## Immunization and nanobody library generation

Antibodies to NDST1 were raised in a llama by intra-muscular immunization with purified protein using Gerbu LQ#3000 as the adjuvant. Immunizations and handling of the llama were performed under the authority of the project license PA1FB163A (University of Reading, UK). Total RNA was extracted from peripheral blood mononuclear cells, and V$_{HH}$ complementary DNAs were generated by RT-PCR. The pool of V$_{HH}$-encoding sequences was amplified by two rounds of nested PCR and cloned into the SfiI sites of the phagemid vector pADL-23c[61]. Electrocompetent *E. coli* TG1 cells were transformed with the recombinant pADL-23c vector, and the resulting TG1 phagemid library stock stored at −80 °C.

## Nanobody panning

The VHH-displaying phage library was recovered by inoculation into 2x TY media supplemented with 100 μg mL$^{-1}$ ampicillin, followed by incubation at 37 °C, 250 rpm until OD$_{600}$ 0.4−0.6. The culture was then infected with M13 helper phage and incubated for an hour at 37 °C, before centrifugation at 2800*g* for 10 min at ambient temperature. The pellet was resuspended in 2x TY media supplemented with 100 μg mL$^{-1}$ ampicillin and 50 μg mL$^{-1}$ kanamycin and grown overnight at 25 °C, 250 rpm with shaking to amplify the library. The amplified VHH-phage

library was recovered by centrifugation at 3200*g* for 15 min at 4 °C, to remove bacterial cells, and the supernatant collected. VHH-presenting phage were subsequently precipitated from the supernatant by the addition of 0.2 volumes of 20% (w/v) PEG6000, 2.5 M NaCl and incubated on ice for 1 h. The precipitated VHH-phage library was collected by centrifugation at 2300*g* for 10 min at 4 °C, then resuspended in 1 mL ice-cold PBS. A further centrifugation step was performed at 20,000*g*, 4 °C for one minute to remove bacterial contaminants. The VHH-phage library was reprecipitated by the addition of 0.2 volumes of 20% PEG6000 (w/v), 2.5 M NaCl and incubated on ice for 30 min, harvested by centrifugation at 20,000*g*, 4 °C for 15 min and resuspended in ice-cold PBS, before being stored at 4 °C.

The VHH-phage library was enriched for NDST1 binders by two rounds of bio-panning. The library was first blocked with StartingBlock (Thermo) for 30 min, followed by an incubation with 50 nM biotinylated NDST1 for an hour, rotating at room temperature. Pre-blocked Dynabeads (Thermo) were mixed with the VHH-phage-biotinylated NDST1 mixture, and incubated for 15 minutes at room temperature. Dynabeads were captured and washed six times with PBS-T, followed by one wash with PBS, before bound phage were released by digestion with 0.25 mg mL$^{-1}$ trypsin (Sigma-Aldrich) in 10 mM Tris, 137 mM NaCl, 1 mM CaCl$_2$, for 30 min at room temperature. The eluted enriched VHH-phage library was collected and amplified by inoculation into exponentially growing TG1 *E. coli* cells, which were grown for 30 min at 37 °C with shaking, then harvested by centrifugation at 2800*g* for 10 min before being resuspended in 1 mL of 2xTY media and plated onto LB agar containing 100 µg mL$^{-1}$ ampicillin. Following incubation overnight at 37 °C, TG1 *E. coli* cells were collected in 2xTY media containing 25% glycerol and stored at −80 °C for further use. The recovered amplified and enriched phage library was used for a further round of bio-panning, performed with 5 nM biotinylated NDST1. Enrichment after bio-panning was determined by plating a 10-fold serial dilution of the recovered cell culture on LB agar plates containing 100 µg mL$^{-1}$ ampicillin.

For screening for NDST1 binding clones, 93 individual clones were selected and grown in 2x TY media (100 µg mL$^{-1}$ ampicillin) overnight at 37 °C, 250 rpm. Overnight cultures were inoculated into 2x TY media containing 100 µg mL$^{-1}$ ampicillin and incubated for 3 h at 37 °C, 250 rpm shaking, before being infected with M13 helper phage. The infected cultures were grown for a further hour at 37 °C, 250 rpm before being centrifuged at 2800*g* for 10 min at 4 °C. The cell pellet was resuspended in 2x TY media containing 100 µg mL$^{-1}$ ampicillin and 50 µg mL$^{-1}$ kanamycin, before being grown at 25 °C, 250 rpm overnight. Clonal VHH-presenting phage were harvested from the culture supernatant by centrifugation at 4000*g* for 15 min at 4 °C to remove bacterial cells, and the supernatant stored at 4 °C for further analysis.

### Enzyme-linked immunosorbent assays to identify NDST1 binding phage
A 96-well high binding microtiter plate (Greiner) was coated with 5 µg mL$^{-1}$ neutravidin in 100 µL PBS per well, and stored overnight at 4 °C. The plate was then washed 5x with PBS-T before the addition of 100 µL of biotinylated NDST1 (50 nM in 20 mM MES, 100 mM NaCl, pH 6.5, 0.1% fat free skimmed milk) per well, and then incubated for 1 h at room temperature, with agitation. The plate was washed 5x with 20 mM MES, 100 mM NaCl, pH 6.5 before being blocked with PBS-T, 2% non-fat skimmed milk (250 µL per well) for 1 h at room temperature, with agitation. Following a further 5 washes with PBS-T, 100 µL of each clonal VHH-presenting phage stock (diluted with 1 volume of PBS-T, 2% non-fat skimmed milk) was added to each well and incubated for 1 h at room temperature, with agitation. The plate was washed five times with PBS-T, before being incubated for 1 h at room temperate with 100 µL anti-M13-HRP (Cytiva 27-9421-01; 1:5000 dilution in StartingBlock; Thermo) per well, with agitation. The plate was washed 5x with PBS-T before the addition of 100 µL per well TMB substrate (SeraCare).

Absorbance was read at 405 nM using a ClarioStar plate reader (software version 5.70 R2).

### Nanobody expression and purification
Identified VHH hits were transformed into WK6 *E. coli* cells using electroporation. 1 L of TB media containing 100 µg mL$^{-1}$ ampicillin, 0.1% glucose (w/v) and 1 mM MgCl$_2$ was inoculated and grown at 37 °C, 250 rpm until OD600 1–1.2. Expression was induced with 1 mM IPTG and the culture incubated at 28 °C overnight, 250 rpm shaking, before harvesting by centrifugation at 4000*g* for 15 min at 4 °C. nAbs were extracted from the periplasm by resuspending the cell pellet in TES buffer (200 mM Tris pH 8.0, 0.5 mM EDTA, 500 mM sucrose) overnight with constant agitation. Two volumes of 50 mM Tris pH 8.0, 125 mM sucrose, supplemented with 120 kU DNase I was added, and the resulting suspension incubated for a further 2 h to lyse cells. Cell debris was removed by centrifugation at 40,000*g* for 30 minutes at 4 °C and the supernatant collected. The supernatant was diluted with 5 volumes of PBS and filtered before being loaded onto a 5 mL HisTrap FF column (Cytiva) pre-equilibrated in Histrap buffer A (PBS, 30 mM imidazole). The column was washed with 5 CV of the same buffer, before elution of the nanobody with a linear gradient of 0-100% Histrap buffer B (PBS, 300 mM imidazole) over 10 CV. Fractions containing nAbs, as judged by SDS-PAGE, were pooled and concentrated using a 10 kDa MWCO Vivaspin filter. nAbs were purified further by SEC using a HiLoad 16/600 Superdex 75 pg column pre-equilibrated in PBS. nAb containing fractions were pooled, concentrated to >2 mg mL$^{-1}$ and flash frozen in LN$_2$. nAbs were stored at −80 °C until required. DNA and protein sequences for nAbs are available in Supplementary Table 2.

### Binding analyses—SPR
Surface plasmon resonance experiments were performed using a Biacore T200 system (GE Healthcare; control software version 2.0.2) primed with 20 mM MES pH 6.5, 100 mM NaCl, 0.05% Tween-20. All assays were performed using a BiotinCAP Sensor chip (GE Healthcare) at 25 °C. To determine the binding affinity of nanobody clones for NDST1, biotinylated NDST1 (10 µM) was immobilized onto the sample channel of the sensor chip at a flow rate of 2 µL min$^{-1}$ for 600 s. The reference channel was left blank. A titration of each nanobody (from 10 µM) was injected over both sensor channels at a flow rate of 30 µL min$^{-1}$, for 100 s. Dissociation was monitored for 100 s. Steady-state binding responses were fitted to a 1:1 binding model (Data evaluation software version 3.1; One-site specific binding) using GraphPad Prism 10.1.0 to fit $K_D$ values. In combinatorial assays, nanobodies were either pre-mixed (1:1, 10 µM) or injected alone at a flow rate of 30 µL min$^{-1}$ for 100 s, with monitoring of the dissociation for 100 s, followed by comparison of the maximal response during the association phase.

### Binding analyses—BLI
Biolayer interferometry (BLI) experiments were performed with an Octet R8 system (Sartorius; software Octet BLI discovery version 12.2.2.20) using Octet SA biosensors. For determination of the binding affinity of nanobodies to NDST1, biosensors were hydrated with two washes of 20 mM MES pH 6.5, 100 mM NaCl, before immobilization of biotinylated NDST1 (75 nM) for 300 s in the same buffer. The loaded sensors were washed twice for 60 s in PBS-T, and the baseline recorded. The association of nanobodies in PBS-T to immobilized NDST1 or unloaded reference sensors were recorded for 100–600 s, followed by measurement of the dissociation in PBS-T for 100–300 s. All steps were performed with shaking at 1000 rpm at 25 °C. Control reactions containing no nanobody and reference sensors were subtracted prior to alignment and smoothing of the sensorgrams with Savitzky-Golay filtering using Octet Analysis studio version 12.2.2.26 (Sartorius). Steady-state binding responses were fitted to a 1:1 binding model (One-site specific binding) using GraphPad Prism 10.1.0 to calculate $K_D$ values.

For determination of the binding affinity of NDST1 to K5, Octet SA biosensors were hydrated with three washes in PBS-T for 300 s before immobilization of biotinylated-K5 (25 µg mL$^{-1}$ or 50 µg mL$^{-1}$) in the same buffer for 300 s. The loaded sensors were washed twice, for 60 s, in PBS-T and the baseline recorded. Association of NDST1 to immobilized biotinylated-K5 or unloaded reference sensors was recorded for 300 s followed by measurement of dissociation in PBS-T for 300 s. All steps were performed with shaking at 1000 rpm at 25 °C. Control reactions containing no NDST1 and reference sensors were subtracted prior to smoothing of the sensorgrams with Savitzky-Golay filtering using Octet Analysis studio version 12.2.2.26. Data were fitted using a 2:1 heterogenous binding model with global fitting in Octet Analysis studio to determine $k_{assoc}$, $k_{dis}$ and $K_D$ values. The effect of nanobodies nAb7 and nAb13 on NDST1 binding to K5 was assessed using the same conditions, with the inclusion of 10 µM nanobody. To evaluate the effect of NaCl on NDST1 binding to K5, an 8 point 1.5-fold dilution of 1.5 M NaCl into 10 mM phosphate buffer pH 7.4, 0.05% Tween-20 was performed using the same assay conditions.

## Differential scanning fluorimetry

Differential scanning fluorimetry experiments were conducted using a QuantStudio5 real-time PCR system running QuantStudio Design & Analysis Software v1.5.2 (Applied Biosystems), employing the TAMRA filter setting, with a step increase of 0.5 °C per 30 s from 25 to 95 °C. The thermal stability of NDST1 (1 µg) in the presence of different nAb concentrations was measured in a total volume of 20 µL in PBS pH 7.4 containing 20X Sypro Orange (Invitrogen). Control reactions containing equivalent amounts of nAb or NDST1 were run alongside on the same plate. Denaturation temperatures ($T_M$) were quantitated from the peak of the first differential of the resulting fluorescence curves, as calculated using Protein Thermal Shift Software v1.4 (Applied Biosystems). Raw data and first differential plots were visualized using GraphPad Prism 10.1.0.

## Generating NDST1 nanobody complexes

NDST1 and nanobodies were incubated in 20 mM MES pH 6.5, 200 mM NaCl, at 1:1 mass ratios for 1 h at room temperature. Complexes were isolated by size exclusion chromatography on a Superdex 200 Increase 10/300 GL column (Cytiva), pre-equilibrated in the same buffer. NDST1-nAb7 and NDST1-nAb13 containing fractions were pooled and concentrated to >5 mg mL$^{-1}$, using a 10 kDa MWCO vivaspin concentrator. Small aliquots were flash frozen using liquid nitrogen and stored at −80 °C for further use, or used immediately.

## Cryo-EM sample preparation

Frozen aliquots of NDST1-nAb7 or NDST1-nAb13 were thawed rapidly and protein concentration adjusted to 0.4–0.6 mg mL$^{-1}$ using 50 mM Tris-HCl pH 7.5, supplemented with PAP (5 mM) and MgCl$_2$ (15 mM). 2.5 µL of samples were applied to Quantifoil R1.2/1.3 300 Cu mesh grids, glow discharged for 60 s at 30 mA negative current in a GloQube Plus glow discharger (Quorum Technologies). Grids were blotted for 1–3 s using a blot force of 3 before plunging into liquid ethane using a Vitrobot Mark IV (Thermo) set to 4 °C, 100% humidity. Grids were clipped into Autogrid rings (Thermo) before loading into microscopes for screening and data collection.

## Cryo-EM data collection and processing

Th NDST1 alone and NDST1-nAb13 datasets were collected from the same cryo-EM grid, in which a large proportion of NDST1-nAb13 complexes had dissociated, enabling the classification of both nAb13 bound and unbound particles. Data was collected on a Krios G4 microscope using an accelerating voltage of 300 kV, Falcon 4i direct electron detector and Selectris X energy filter with a slit width of 10 eV. Movies were collected using EPU (version 3.2.0) in EER format at 165,000x magnification, using a total dose of 50 electrons per Å$^2$, a calibrated pixel size of 0.73 Å, and target defocus values of −1.2 µm to −2.6 µm in 0.2 µm intervals.

The NDST1-nAb7 dataset was collected on a Krios G3i microscope, using an accelerating voltage of 300 kV, Falcon 4 direct electron detector and Selectris X energy filter with a silt width of 10 eV. Movies were collected using EPU (version 3.2.0) in EER format at 165,000x magnification, using a total dose of 50 electrons per Å$^2$, a calibrated pixel size of 0.73 Å, and target defocus values of −1.2 µm to −2.6 µm in 0.2 µm intervals.

EM processing workflows for each dataset are outlined in Supplementary Fig. 7. In brief, micrograph pre-processing was performed in cryoSPARC version 2.14.1[62–64] live interface using path-motion CTF correction. Initial particles were picked using unbiased blob-picking, and good particles isolated via two rounds of 2D classification. 2D class averages with secondary structure features were used to guide template-based particle picking, followed by further rounds of 2D classification. Ab initio models generated in cryoSPARC were used as non-biased references for 1 or more rounds of heterogenous refinement.

The best classes from heterogenous refinement, as judged by resolution, were pooled and refined further using cryoSPARC non-uniform refinement, followed by rounds of local and global CTF refinement where appropriate, followed again by non-uniform refinement. Final postprocessing and local resolution estimates were carried out in Relion 3.1.1[65]. 3DFSC calculations were carried out using the Remote 3DFSC Processing Server (3dfsc.salk.edu)[66].

3D-variability analyses on the NDST1-nAb7 and NDST1-nAb13 complexes were each carried out using the cryoSPARC pipeline, incorporating 3 variability components. Focussed local refinements for both complexes were carried out in cryoSPARC, with masking to subtract either the NTD or the sulfotransferase domain, generating subtracted particles for the deacetylase-sulfotransferase-nAb or NTD-deacetylase-nAb regions respectively. Locally refined volumes were postprocessed using Relion 3.1.1, before being combined using UCSF ChimeraX, by taking the maximum value from either locally refined map at each voxel. The composite map was used to guide model building. All other analyses and depositions were carried out with the individual locally refined maps.

For all structures, cryo-EM model building was carried out in COOT[67], using starting models initially derived from AlphaFold2 (NDST1) or the PDB (for nAb7, from accession 7TGF). Models were improved by manual building using COOT, iterated with rounds of real-space refinement using PHENIX.REFINE[68]. The nAb13 nanobody structure was derived from the nAb7 structure by conversion to an all Cα chain in COOT, and removal of the long nAb7 unique CDR3 loop (R98–F116), before fitting the Cα model into the NDST1-nAb13 volume by 5 rounds of rigid-body refinement in PHENIX.REFINE. Structural figures were generated using ChimeraX[69]. Electrostatic surfaces were calculated using ChimeraX. Binding interactions were determined using PISA analysis[70] with default settings. RMSD values were calculated using CCP4MG[71]. Sequence alignment values were calculated using ClustalΩ[72]. All cryo-EM processing and model building statistics are shown in Supplementary Table 1.

## Molecular docking

Interactions between NDST1 and HS were analyzed by molecular docking using GlycoTorch Tools and GlycoTorch Vina (GTV)[41]. The protonation of residues of both protein and ligands were set to reflect physiological pH 7.4. Glycam carbohydrate builder (Glycam.org) was used to build the two octasaccharides, [GlcNAc-GlcA]$_4$ and [GlcNAc-GlcA]$_2$-[GlcNS-GlcA]$_2$. GTV was used to convert the octasaccharides obtained from Glycam to pdbqt format. A centroid selected from the PAP binding site and metal binding sites from the cryo-EM structure was used to define the coordinates for the docking of heparosan oligosaccharides. Docking of oligosaccharides was performed using the

grid box sizes of 40 × 50 × 40 Å and 55 × 40 × 25 Å from the centroids in the deacetylase and sulfotransferase domains respectively. The octasaccharides were docked using GTV with energy range, exhaustiveness and num_modes values set to be 12, 80, and 100, respectively. The conformations chosen for further analysis were based on the best-estimated affinities and their ranking in the clustering process. Docking visualization and figures were generated using ChimeraX[69].

## Reporting summary

Further information on research design is available in the Nature Portfolio Reporting Summary linked to this article.

## Data availability

The cryo-EM data generated in this study have been deposited in the PDB and EMDB under accession codes 8CCY (NDST1 alone model), EMD-16564 (NDST1 alone map), 8CD0 (NDST1-nAb7 model), EMD-16627 (NDST1-nAb7 NTD and deacetylase domain local map), EMD-16629 (NDST1-nAb7 deacetylase and sulfotransferase domain local map), EMD-16565 (NDST1-nAb7 composite map), EMD-16626 (NDST1-nAb7 original map), 8CHS (NDST1-nAb13 model), EMD-16662 (NDST1-nAb13 deacetylase and sulfotransferase domain local map), EMD-16663 (NDST1-nAb13 deacetylase and sulfotransferase domain local map), EMD-16664 (NDST1-nAb13 composite map), EMD-16661 (NDST1-nAb13 original map). We have also referenced in this work PDB accessions 6NWZ (Agd3 deacetylase), 3UAN (HS3ST1), 6XL8 (HS3ST3), and EMDB accession EMD-17349 (NDST1 dimer). DNA and protein sequences for NDST1 and nAb constructs are available in Supplementary Table 2. Source data are provided with this paper.

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

## Acknowledgements

This work was supported by the Rosalind Franklin Institute, with funding delivery partner the Engineering and Physical Sciences Research Council (EPSRC) UK. We acknowledge support from the Biotechnology and Biological Sciences Research Council, UK (BB/L023717/1 & BB/X011739/1 to MAS; BB/V018523/1 to RJO), the Engineering and Physical Sciences Research Council, UK (EP/X019179/1 to MAS), and the Wellcome Trust (Sir Henry Dale Fellowship 218579/Z/19/Z to LW). Electron microscopy provision was provided via the Rosalind Franklin institute, and through the Oxford particle Imaging Centre (OPIC) facility, a UK Instruct-ERIC Centre founded by a Wellcome JIF award (060208/Z/00/Z) and supported by a Wellcome equipment grant (093305/Z/10/Z). Financial support for HMED was provided by the Wellcome Trust Core Award Grant (203141/Z/16/Z). Computation for cryo-EM analysis was performed using the Oxford Biomedical Research Computing (BMRC) facility, a joint development between the Wellcome Centre for Human Genetics and the Big Data Institute supported by Health Data Research

UK and the NIHR Oxford Biomedical Research Centre. NSG is supported by QUT Accelerate Fellowship. Computational (and/or data visualization) resources and services used in this work were provided by the eResearch Office, Queensland University of Technology, Brisbane, Australia. NSG would also like to acknowledge the National Computational Infrastructure (NCI), Australia for providing the high-performance computing facility to carry out molecular docking simulations. We thank Jiandong Huo for assistance with phage display library construction, and Mark Arrowsmith at the University of Keele School of Pharmacy and Bioengineering for assistance with ICP-OES.

## Author contributions

L.W. and C.J.M.W. conceived and designed experiments. L.W. and C.J.M.W. carried out protein expression and purification. C.J.M.W. carried out polysaccharide production and purification, with assistance from M.A.S. C.J.M.W. and S.A. generated nanobodies, under supervision from R.J.O. C.J.M.W. carried out biochemical and biophysical analyses. C.J.M.W., H.M.E.D., and L.W. carried out cryo-EM sample preparation, data collection and data analysis. N.S.G. carried out molecular docking simulations. L.W. wrote the manuscript with input from all authors.

## Competing interests

The authors declare no competing interests.
