## [Peer Review File · Nature Communications]

Structural and mechanistic characterization of bifunctional heparan sulfate N-deacetylase-N-sulfotransferase 1REVIEWER COMMENTS

Reviewer #1 (Remarks to the Author):

This manuscript reports the structure of NDST1, a ~75 kDa enzyme with no symmetry, in complex with a nanobody at impressive resolution of 0.24, a second structure of the unliganded enzyme to 0.27 nm, and a third one to a more modest resolution of 0.32 nm. Surprisingly, the half-maps for the partial structures were not deposited in the EMDB and, the only FSC plot was made available for the 0.27 nm structure. The authors do present an orientation plot showing varying degrees of preferred orientation. The maps submitted for review are consistent with resolution claims. However, FSC plots are important to evaluate the quality of the reconstructions and the absence of artifacts. The half maps should be deposited in the EMDB/PDB database for all entries. Further, I would recommend the addition of a panel showing the results of 3DFSC.

Reviewer #2 (Remarks to the Author):

The manuscript by Mycroft-West et al. describes the cryo-EM structure of the heparan sulfate N-deacetylase-N-sulfotransferase 1. This is a bifunctional enzyme that catalyzes the deacetylation of N-acetylglucosamine and its subsequent sulfation in order to generate glucosamine-N-sulfate. Being the first step of heparan sulfate chain maturation, it is crucial for the generation of highly sulfated domains that in turn represent essential motifs to be recognized by diverse protein interactors. Most enzymes in the heparan sulfate biosynthesis pathway have been structurally characterized over the last two decades with the NDST family remaining one of the last enzymes resisting mechanistic characterization. The article presents the purification of soluble NDST1 and the generation of activity-modulating nanobodies. Further, three high-resolution structures of NDST1 in presence/absence of nanobodies are described. Despite a previous crystal structure of the sulfotransferase domain (Kakuta et al. 1999) and a very recent intermediate resolution cryo-EM structure of the bifunctional enzyme (Vallet et al., 2023) these high-resolution structures provide important new mechanistic insights and are of high interest to the glycobiology community. The work presented is technically excellent, very well written and fits very well the scope of Nature communications.

Some minor issues should be addressed:

-P8 L191-192: Binding affinities of nanobodies range from 120 nM to 2.67 μ M and thus, to be more precise, span the "high" nanomolar to "low" micromolar range.

-P8 L199-L200: nAb13 binding increases NDST1 activity. It would be very interesting to see if this nanobody has a more pronounced effect on thermal stability compared to nAb5 and nAb7.

-P11 L230: "with the resulting map of sufficient quality to model of nearly all residues". There is something wrong in this sentence.

- Figure 4: the nAb13-bound structure seems to be in a more closed conformation than the nAb7-bound NDST1 structure. Are there differences in the angle of the hinge region between nAb-bound and unbound structures?

- Figure S6: Please provide FSC curves for final maps (masked and unmasked).

- Figure S6 B: While overall resolution of local maps is proposed to be at 2.42 Å, the local resolution illustration starts at 2.5 Å and thus not in the relevant range.

- P25 L566-569: multimer formation was also shown for NDST1 (Vallet et al., 2023). It should be discussed more carefully, why no oligomer formation is observed in this study.

- discuss the potential influence of DTT used during purification on correct structure of NDST1, which is proposed to harbor a disulfide-bridge (C818-C828) in its N-terminal domain.

Reviewer #3 (Remarks to the Author):

In this manuscript, cryo-electron microscopy has been used to determine the structure of human N-deacetylase/N-sulfotransferase 1 (NDST1). In addition, activity modulating nanobodies have been used to elucidate potential cooperativity between the N-deacetylase and N-sulfotransferase domains. Critical residues important for N-deacetylation have also been identified by structural comparisons with the fungal deacetylase Agd3 from *Aspergillus fumigatus*, confirmed by studies of generated mutants.

Major point

1. The structure of the NDST1, a key enzyme in heparan sulfate biosynthesis has long been awaited. Two labs have been working on this in parallel and recently, the study of Vallet et al. was published in *Proteoglycan Research*. While some of the data from the two labs of course are overlapping, unique results are presented in both articles. Therefore, the results presented and discussed in Vallet et al. need to be commented on in the present manuscript.

In general, the present manuscript is stronger on the structural cryo-EM data (resolution of 2.7 Å, compared to 4.5 Å in the study by Vallet et al.). The use of nanobodies, a more thorough characterization of the N-deacetylase active site and the assignment of Ca²⁺ as the metal cofactor (Fig. S2) are also merits of the present manuscript. However, the finding of Vallet et al. that NDST1 to a great extent occurs as dimers, and the tight coupling of N-deacetylation and N-sulfation need to be discussed in relation to the findings of the present manuscript. Now, the Vallet paper is only mentioned briefly in the Discussion (ref. 41). How does the “dimer theory” fit with your data?

Other points that need to be discussed is the role of the “third domain”, where the present manuscript shows interesting data pointing to a role for this domain in controlling enzyme substrate interactions. Another important finding of the present manuscript is that binding of the N-sulfotransferase domain to the substrate may happen before N-deacetylation. How does this fit with the dimer theory (Fig. S14)?

Minor points

1. Abstract: I would add “mainly” in the sentence “We have determined cryo-electron microscopy structures of human N-deacetylase-N-sulfotransferase (NDST)1, the bifunctional enzyme mainly responsible for initial GlcNS modification of HS.”

2. Introduction, l. 63: The glycosaminoglycan family also contains keratan sulfate which instead of hexuronic acid contains galactose.

3. Introduction, l.72: The modification reactions take place at the same time as the chain is growing.

4. Introduction, l. 95: It should also be mentioned that loss of Ndst1 also causes a lung phenotype (Fan et al. (2000) FEBS Lett.; Ringvall et al. (2000) J. Biol. Chem.)

5. Results, l. 133: It should be mentioned that expression of NDST1 in insect cells could influence N-glycosylation which in turn could affect enzyme activity.

6. Results, l. 137: Since N-deacetylation is the limiting enzyme activity of NDST1 (see ref. 18), the enzyme assay used actually measures this activity.

7. Results, l. 435: The salt sensitivity of NDST1 was not shown in ref. 39. (Don't remember where it was first shown for NDST1).

Reviewer #4 (Remarks to the Author):

In this manuscript the authors describe the structure of human N-deacetylase-N5 sulfotransferase 1 (NDST1) and propose its catalytic mechanism, which entails a ratchet-like process where the ST domain initiates binding based on electrostatic interactions. An atomistic model of the octasaccharide recognition is presented based on molecular docking calculations, which I have been asked to assess.

In my opinion, molecular docking can be useful to indicate the location of the binding site, which indeed closely resembles the one in the PDBs of other NDSTs as specified in Supplementary Material, yet it is not an ideal technique to provide any further insight, such as sequence specificity or mechanisms of recognition, especially in such a complex system.

From my reading the use of the molecular docking results in this work is exactly to provide an atomistic 'picture' of the bound HS, nothing more, so I do think that within this scope it is a very useful addition to the work. As a note, the generation of a molecular ensemble of structures of the substrate octasaccharide may have been very useful to expand the flexibility of the substrate and obtain better agreement with the PDB structures and maybe even a better conformation around the PAP ligand (p17 line 390).

Responses to reviewer comments

- Reviewer #1 (Remarks to the Author):

This manuscript reports the structure of NDST1, a ~75 kDa enzyme with no symmetry, in complex with a nanobody at impressive resolution of 0.24, a second structure of the unliganded enzyme to 0.27 nm, and a third one to a more modest resolution of 0.32 nm. Surprisingly, the half-maps for the partial structures were not deposited in the EMDB and, the only FSC plot was made available for the 0.27 nm structure. The authors do present an orientation plot showing varying degrees of preferred orientation. The maps submitted for review are consistent with resolution claims. However, FSC plots are important to evaluate the quality of the reconstructions and the absence of artifacts. The half maps should be deposited in the EMDB/PDB database for all entries. Further, I would recommend the addition of a panel showing the results of 3DFSC.

We apologize for this omission. The half maps for the locally refined volumes were actually deposited in the EMDB (see updated data availability statement). However, only the validation reports for the model + composite maps were previously provided for peer review (our fault). We have now provided a full set of validation reports, including locally refined volumes, for the reviewers' attention. Coordinates and maps are also available to reviewers on request.

We have updated the data availability statement to clarify what has been deposited.

(P28L695): "Cryo-EM coordinates have been deposited in the PDB and EMDB under accession codes 8CCY (NDST1 alone model), EMD-16564 (NDST1 alone map), 8CD0 (NDST1-nAb7 model), EMD-16627 (NDST1-nAb7 NTD and deacetylase domain local map), EMD-16629 (NDST1-nAb7 deacetylase and sulfotransferase domain local map), EMD-16565 (NDST1-nAb7 composite map), EMD-16626 (NDST1-nAb7 original map), 8CHS (NDST1-nAb13 model), EMD-16662 (NDST1-nAb13 deacetylase and sulfotransferase domain local map), EMD-16663 (NDST1-nAb13 deacetylase and sulfotransferase domain local map), EMD-16664 (NDST1-nAb13 composite map), EMD-16661 (NDST1-nAb13 original map)."

FSC and 3DFSC curves have been added to Fig S7 (used to be S6) as requested.

- Reviewer #2 (Remarks to the Author):

The manuscript by Mycroft-West et al. describes the cryo-EM structure of the heparan sulfate N-deacetylase-N-sulfotransferase 1. This is a bifunctional enzyme that catalyzes the deacetylation of N-acetylglucosamine and its subsequent sulfation in order to generate glucosamine-N-sulfate. Being the first step of heparan sulfate chain maturation, it is crucial for the generation of highly sulfated domains that in turn represent essential motifs to be recognized by diverse protein interactors. Most enzymes in the heparan sulfate biosynthesis pathway have been structurally characterized over the last two decades with the NDST family remaining one of the last enzymes resisting mechanistic characterization. The article presents the purification of soluble NDST1 and the generation of activity-modulating nanobodies. Further, three high-resolution structures of NDST1 in presence/absence of nanobodies are described. Despite a previous crystal structure of the sulfotransferase domain (Kakuta et al. 1999) and a very recent intermediate resolution cryo-EM structure of the bifunctional enzyme (Vallet et al., 2023) these high-resolution structures provide important new mechanistic insights and are of high interest to the glycobiology community. The work presented is technically excellent, very well written and fits very well the scope of Nature communications.

Some minor issues should be addressed:

-P8 L191-192: Binding affinities of nanobodies range from 120 nM to 2.67 μM and thus, to be more precise, span the "high" nanomolar to "low" micromolar range.

Fixed

(P8L199): "Binding affinities (KDs) spanned the high nanomolar to low micromolar range..."

-P8 L199-L200: nAb13 binding increases NDST1 activity. It would be very interesting to see if this nanobody has a more pronounced effect on thermal stability compared to nAb5 and nAb7.

We conducted differential scanning fluorimetry experiments to assess the thermal stability of NDST1 in the presence of nAb5, nAb7

and nAb13. In summary, we see stabilization in the presence of all nanobodies, with greatest stabilization for the nAb7 complex, broadly in keeping with the higher NDST1 affinity of this nAb vs nAb5 and nAb13. The text has been updated with the DSF results, and a new Figure S5 has been added to the SI.

(P8L209): “Differential scanning fluorimetry showed ~2.6°C, ~4.4°C and ~1.9°C stabilization of NDST1 thermal denaturation in the presence of 10.8 μM nAb5, nAb7 and nAb13 respectively, broadly in line with their relative binding affinities for the enzyme (Figure S5).”

-P11 L230: *"with the resulting map of sufficient quality to model of nearly all residues". There is something wrong in this sentence.*

Fixed

(P11L243): “...with the resulting map of sufficient quality to model nearly all residues along the protein chain”

Figure 4: the nAb13-bound structure seems to be in a more closed conformation than the nAb7-bound NDST1 structure. Are there differences in the angle of the hinge region between nAb-bound and unbound structures?

The reviewer is correct there is a slightly more closed conformation for the nAb13 complex (also for the nAb7 complex) compared to nAb free NDST1. However, our opinion is that these primarily reflect the flexibility of the NDST1-nAb complexes (see Supplemental Videos), with insufficient evidence that any conformational changes are stabilized by nAb binding. Crucially, the closed NDST1 conformations become apparent after combining partial maps from local refinement. Thus the final ‘hinge’ angle in the composite map may simply arise from the orientation adopted by the partial maps during local refinement processing.

We have amended the text, and added Figure S14, which we hope covers the hinge angle differences in a manner without overinterpreting:

(P14L333) “The overall structure of NDST1 complexed to nAb7 was similar to that of free NDST1, except for a slightly more ‘closed’ conformation of the NDST1 ‘elbow’, consistent with conformational flexibility around its hinge (Figure S14; Supplementary Movie 1).”

(P15L349) “As with nAb7, the nAb13 complex displayed a more ‘closed’ NDST1 conformation, as well as improved ordering of the D319–V332, C486–G512 and T528–L538 loops (Figure S14; Figure S15).”

- *Figure S6: Please provide FSC curves for final maps (masked and unmasked).*

FSC and 3DFSC curves have been added to Figure S7 (used to be S6). See also response to Reviewer 1.

- *Figure S6 B: While overall resolution of local maps is proposed to be at 2.42 Å, the local resolution illustration starts at 2.5 Å and thus not in the relevant range.*

Local resolution maps in Figure S7B now span 2.4–3.1 Å.

- *P25 L566-569: multimer formation was also shown for NDST1 (Vallet et al., 2023). It should be discussed more carefully, why no oligomer formation is observed in this study.*

We believe the lack of oligomerization in our work compared to Vallet *et al* reflect differences in either our construct design (6xHis tagged vs phosphatase tagged in Vallet *et al*), or expression system (insect vs mammalian in Vallet *et al*). These caveats have been added to the text, along with an extended discussion upon the mechanistic implications of NDST1 multimerization:

(P25L586) “Whilst stable NDST1 oligomers were not observed in this work, possibly due to differences in construct design (6xHis-tagged vs phosphatase-tagged) or expression system (insect vs mammalian) compared to Vallet et al...”

See also response to Reviewer 3.

- *discuss the potential influence of DTT used during purification on correct structure of NDST1, which is proposed to harbor a disulfide-bridge (C818-C828) in its N-terminal domain.*

We have added Figure S8 and a short discussion on disulfides to the text:

(P11L249) “Disulfide bonds between C586–C601 and C818–C828 were clearly visible in the cryo-EM density, with the former at lower occupancy, possibly due to its surface location being susceptible to reduction by 1 mM DTT, present in the sample buffer (Figure S8).”

- **Reviewer #3 (Remarks to the Author):**

*In this manuscript, cryo-electron microscopy has been used to determine the structure of human N-deacetylase/N-sulfotransferase 1 (NDST1). In addition, activity modulating nanobodies have been used to elucidate potential cooperativity between the N-deacetylase and N-sulfotransferase domains. Critical residues important for N-deacetylation have also been identified by structural comparisons with the fungal deacetylase Agd3 from *Aspergillus fumigatus*, confirmed by studies of generated mutants.*

Major point

*1. The structure of the NDST1, a key enzyme in heparan sulfate biosynthesis has long been awaited. Two labs have been working on this in parallel and recently, the study of Vallet et al. was published in *Proteoglycan Research*. While some of the data from the two labs of course are overlapping, unique results are presented in both articles. Therefore, the results presented and discussed in Vallet et al. need to be commented on in the present manuscript.*

In general, the present manuscript is stronger on the structural cryo-EM data (resolution of 2.7 Å, compared to 4.5 Å in the study by Vallet et al.). The use of nanobodies, a more thorough characterization of the N-deacetylase active site and the assignment of Ca²⁺ as the metal cofactor (Fig. S2) are also merits of the present manuscript. However, the finding of Vallet et al. that NDST1 to a great extent occurs as dimers, and the tight coupling of N-deacetylation and N-sulfation need to be discussed in relation to the findings of the present manuscript. Now, the Vallet paper is only mentioned briefly in the Discussion (ref. 41). How does the “dimer theory” fit with your data?

We apologise for the clear omission, and wholeheartedly agree a more detailed discussion of the Vallet paper is appropriate. We have added an extended paragraph in the discussion that covers the potential implications of dimer formation as observed by Vallet et al for our results:

(P24L573) “The precise nature of HS coordination between the NDST1 deacetylase and sulfotransferase domains remains to be elucidated. In the absence of direct visualization of NDST1-HS binding, two models may be considered for how a polysaccharide chain can access both NDST1 catalytic sites in a coordinated fashion. A 1:1 model, in which a single HS chain ‘loops’ around NDST1 to access both catalytic domains, appears somewhat implausible, given the entropic penalty that would arise from constraining extended HS polysaccharides (Figure S17; Figure S19). Alternatively, a 2:1 (or >2:1) model, in which the catalytic cycle is shared between multiple NDST1 enzymes, might operate. Such a model accords well with recent observations of catalytically competent NDST1 homodimers by Vallet et al, in which an alternative arrangement of NDST1 catalytic domains is created by the dimeric interface, although the active sites remain some 60 Å apart (Figure S18)⁴⁶. Formation of NDST1 oligomers may provide a route to transfer HS chains, initially anchored to the sulfotransferase domain of one protomer, to the deacetylase domain of a neighbor, thereby enabling functional coupling in a manner distinct from the 1:1 model (Figure S19). Whilst stable NDST1 oligomers were not observed in this work, possibly due to differences in construct design (6xHis-tagged vs phosphatase-tagged) or expression system (insect vs mammalian) compared to Vallet et al, transient oligomerization may still operate in our system to coordinate NDST1 activities during catalysis. Interestingly, low (~7.3) μM KMs were measured by Vallet et al for processing of heparosan by partially dimeric NDST1, contrasting with the high μM values recorded here (Figure 1c), and consistent with improved functionality for a dimer (albeit caution must be exercised when comparing values measured using different assays). We also do not discount the possibility of further oligomeric NDST1 states within intracellular milieu, formed by the preorganization of enzymes on Golgi membranes. Detailed analysis of NDST1-HS interactions across multiple polysaccharide sequences and NDST1 oligomerization states will be required to dissect the consequences of different substrate binding modes and stoichiometries on catalysis.”

Other points that need to be discussed is the role of the “third domain”, where the present manuscript shows interesting data pointing to a role for this domain in controlling enzyme substrate interactions. Another important finding of the present manuscript is that binding of the N-sulfotransferase domain to the substrate may happen before N-deacetylation. How does this fit with the dimer theory (Fig. S14)?

An extended discussion on the NDST1 NTD has been added:

(P23L538) “Computational docking with HS octasaccharides suggests that the hitherto undescribed NDST1 NTD may play a role in polysaccharide engagement, by contributing to the formation of a substrate binding cleft adjacent to the deacetylase domain. The presence of carbohydrate binding module (CBM) domains that assist substrate binding is common amongst polysaccharide processing enzymes⁴⁵. Indeed, the fungal deacetylase Agd3, which is homologous to the NDST1 deacetylase domain, possesses an N-terminal CBM that enhances substrate attachment³⁸. Whilst this N-terminal domain of Agd3 extends a linear substrate cleft projecting from its catalytic site, the cleft formed by the analogous NDST1 NTD is kinked with respect to the deacetylase domain (Figure S11e), potentially reducing its effectiveness in aiding HS binding. Accordingly, we find that initial NDST1 engagement of HS is driven by interactions at its sulfotransferase, rather than deacetylase domain, despite the latter operating first in the catalytic cycle. A further role of the NDST1 NTD may be in enzyme oligomerization. Vallet et al have recently reported the intriguing observation of stable NDST1 homodimers⁴⁶, which interact via a C2 symmetric interface involving the NTD and sulfotransferase domains (Figure S18). NDST1 oligomerization may be mechanistically significant during catalytic turnover (see below), thus implicating the NTD in functions beyond simple substrate binding.”

Minor points

1. Abstract: I would add “mainly” in the sentence “We have determined cryo-electron microscopy structures of human N-deacetylase-N-sulfotransferase (NDST)1, the bifunctional enzyme mainly responsible for initial GlcNS modification of HS.”

Amended – primarily instead of mainly:

(P2L44) “We have determined cryo-electron microscopy structures of human N-deacetylase-N-sulfotransferase (NDST)1, the bifunctional enzyme primarily responsible for initial GlcNS modification of HS.”

2. Introduction, l. 63: *The glycosaminoglycan family also contains keratan sulfate which instead of hexuronic acid contains galactose.*

Amended:

(P2L63) “HS is a member of the glycosaminoglycan (GAG) family – linear polysaccharides typically comprised of alternating hexosamine and uronic acid monosaccharides (heparin, HS, chondroitin sulfate, dermatan sulfate and hyaluronan), whilst keratan sulfate is comprised of alternating hexosamine and galactose units.”

3. Introduction, l.72: *The modification reactions take place at the same time as the chain is growing.*

Amended:

(P3L75) “Complexity is introduced by a series of modification reactions upon the growing heparosan chain...”

4. Introduction, l. 95: *It should also be mentioned that loss of Ndst1 also causes a lung phenotype (Fan et al. (2000) FEBS Lett.; Ringvall et al. (2000) J. Biol. Chem.)*

Amended and suggested references added:

(P3L98) “Total loss of Ndst1 produces perinatal lethality in mice due to severe cerebral and craniofacial defects²⁷, with impaired lung development also noted^{28,29}.”

5. Results, l. 133: *It should be mentioned that expression of NDST1 in insect cells could influence N-glycosylation which in turn could affect enzyme activity.*

We have added some discussion and Figure S9 to address this fact:

(P11L252) “Proteins produced by baculoviral expression can sometimes differ in glycosylation compared to mammalian expression³⁷. However, we were unable to resolve distinct glycoforms in our NDST1 sample by cryo-EM. Density consistent with the first GlcNAc of an N-glycan tree could be seen at the predicted N-glycosylation site N401, but this was not of sufficient quality to model (Figure S9).”

6. Results, l. 137: *Since N-deacetylation is the limiting enzyme activity of NDST1 (see ref. 18), the enzyme assay used actually measures this activity.*

The reviewer is broadly correct, however the exact rates of each step in the NDST1 catalytic cycle will also depend on local substrate levels. Therefore deacetylation may not be rate limiting under conditions where e.g. PAPS is at a low concentration. We have added the following to the text:

(P5L144) “As deacetylation by NDST1 is typically rate limiting²⁵, depending on substrate concentrations, this coupled enzyme assay primarily probes this step of turnover.”

7. Results, l. 435: *The salt sensitivity of NDST1 was not shown in ref. 39. (Don't remember where it was first shown for NDST1).*

The old Ref 39 (now Ref 42; Wei et al 1993) shows loss of both NDST1 deacetylase and sulfotransferase activity at higher salt concentrations (Fig 3A in that paper). We have also added a new Ref 43 (Riesenfeld et al 1980), which is an earlier demonstration of NDST1 deacetylation activity being sensitive to salt (Fig 5 in that paper).

- Reviewer #4 (Remarks to the Author):

In this manuscript the authors describe the structure of human N-deacetylase-N5 sulfotransferase 1 (NDST1) and propose its catalytic mechanism, which entails a ratchet-like process where the ST domain initiates binding based on electrostatic interactions. An atomistic

model of the octasaccharide recognition is presented based on molecular docking calculations, which I have been asked to assess.

In my opinion, molecular docking can be useful to indicate the location of the binding site, which indeed closely resembles the one in the PDBs of other NDSTs as specified in Supplementary Material, yet it is not an ideal technique to provide any further insight, such as sequence specificity or mechanisms of recognition, especially in such a complex system.

From my reading the use of the molecular docking results in this work is exactly to provide an atomistic 'picture' of the bound HS, nothing more, so I do think that within this scope it is a very useful addition to the work. As a note, the generation of a molecular ensemble of structures of the substrate octasaccharide may have been very useful to expand the flexibility of the substrate and obtain better agreement with the PDB structures and maybe even a better conformation around the PAP ligand (p17 line 390).

We thank the reviewer for their assessment of our docking work. Understanding the precise enzyme substrate interactions involved in NDST1 catalysis is a clear next goal for the study of this enzyme.

Our opinion is that additional docking calculations towards understanding substrate interactions should be accompanied by experimental structures, to gain the kind of mechanistic binding information the reviewer identifies as lacking here. We thereby propose that further ensemble docking is out of scope for this current manuscript, and would be best conducted in conjunction with experimental structure determination in a future project.

Other amendments

1) A brief methods section has been added to the main text. Given their length, full detailed methods and protocols remain in the SI.

2) Table S1 has been updated to include statistics for the original NDST1-nAb7 and NDST1-nAb13 maps before local refinement. Validation reports are provided for the reviewers' attention. The relevant maps are also available upon request.

3) Error ranges have been added to kinetic/binding constants in Fig 1c, Table 1, Fig 4f, Table 2, Figure S2.

4) We have reworded the sentence regarding assignment of deacetylase metal center, to clarify the lack of a complete coordination sphere for the modelled Ca²⁺. We believe this incomplete sphere most likely reflects local disorder in the cryo-EM data, rather than a true biochemical effect (which would be extremely unusual).

(P12L276) "Clear density, distinct from surrounding protein sidechains, was visible at the center of this triad, consistent with Ca²⁺ coordinated to H389, H393, D320, although a full coordination sphere could not be resolved, possibly due to local disorder within the open catalytic cleft (Figure 3e)."

5) We have rerefined our nAb free NDST1 model, to fix a small loop (P665–D671) that was poorly fitted to the density in the original structure. Other aspects of the structure remain the same. Updated validation reports are provided for the reviewers' attention. The updated coordinates are also available upon request. Docking calculations have been rerun on the latest iteration of the NDST1 model, and show essentially the same results as before. All relevant structural figures in the manuscript and SI have been updated with the latest models.

Editorial requests

1) Editorial policy and reporting summary checklists have been provided.

2) Sequences for the NDST1 constructs and NDST1 binding nanobodies used in this work have been included in the Supplementary Information in **Table S2**.

3) Source data for charts and graphs in the figures have been provided.

4) Bar graphs in **Figure 3** and **Figure S2** have been revised to show individual datapoints, rather than average values.

5) Data availability statement has been amended to clarify the PDB and EMD codes, and to mention the inclusion of source data and protein sequences.

REVIEWERS' COMMENTS

Reviewer #1 (Remarks to the Author):

The authors provided all the information requested. I believe this manuscript is adequate for publication.

Reviewer #2 (Remarks to the Author):

All concerns were addressed appropriately.